# The Role of Membrane Transporters in Plant Growth and Development, and Abiotic Stress Tolerance

**DOI:** 10.3390/ijms222312792

**Published:** 2021-11-26

**Authors:** Rafaqat Ali Gill, Sunny Ahmar, Basharat Ali, Muhammad Hamzah Saleem, Muhammad Umar Khan, Weijun Zhou, Shengyi Liu

**Affiliations:** 1Key Laboratory of Biology and Genetic Improvement of Oil Crops, The Ministry of Agriculture and Rural Affairs, Oil Crops Research Institute of Chinese Academy of Agricultural Sciences, Wuhan 430062, China; liusy@oilcrops.cn; 2College of Plant Science and Technology, Huazhong Agricultural University, Wuhan 430070, China; sunnyahmar13@gmail.com (S.A.); saleemhamza312@webmail.hzau.edu.cn (M.H.S.); 3Department of Agronomy, University of Agriculture, Faisalabad 38040, Pakistan; bali@uaf.edu.pk; 4Key Laboratory of Crop Ecology and Molecular Physiology, Fujian Agriculture and Forestry University, Fuzhou 350002, China; umar.khan018@yahoo.com; 5Institute of Crop Science, The Ministry of Agriculture and Rural Affairs Key Laboratory of Spectroscopy Sensing, Zhejiang University, Hangzhou 310058, China; wjzhou@zju.edu.cn

**Keywords:** abiotic stresses, gene expression, genomics, ion homeostasis, plant growth and development, plasma membrane, sugar translocation

## Abstract

The proteins of membrane transporters (MTs) are embedded within membrane-bounded organelles and are the prime targets for improvements in the efficiency of water and nutrient transportation. Their function is to maintain cellular homeostasis by controlling ionic movements across cellular channels from roots to upper plant parts, xylem loading and remobilization of sugar molecules from photosynthesis tissues in the leaf (source) to roots, stem and seeds (sink) via phloem loading. The plant’s entire source-to-sink relationship is regulated by multiple transporting proteins in a highly sophisticated manner and driven based on different stages of plant growth and development (PG&D) and environmental changes. The MTs play a pivotal role in PG&D in terms of increased plant height, branches/tiller numbers, enhanced numbers, length and filled panicles per plant, seed yield and grain quality. Dynamic climatic changes disturbed ionic balance (salt, drought and heavy metals) and sugar supply (cold and heat stress) in plants. Due to poor selectivity, some of the MTs also uptake toxic elements in roots negatively impact PG&D and are later on also exported to upper parts where they deteriorate grain quality. As an adaptive strategy, in response to salt and heavy metals, plants activate plasma membranes and vacuolar membrane-localized MTs that export toxic elements into vacuole and also translocate in the root’s tips and shoot. However, in case of drought, cold and heat stresses, MTs increased water and sugar supplies to all organs. In this review, we mainly review recent literature from Arabidopsis, halophytes and major field crops such as rice, wheat, maize and oilseed rape in order to argue the global role of MTs in PG&D, and abiotic stress tolerance. We also discussed gene expression level changes and genomic variations within a species as well as within a family in response to developmental and environmental cues.

## 1. Introduction

With the current pace, global population is expected to grow with minimum 25% and reach “10 Billions” in 2050 [1]. To combat food insecurity and to feed the growing population, there is need to design studies on whole genome membrane transporters (MTs) for the proper utilization of plant energy under dynamic environmental changes in order to produce high yielding crops. Recent advances evidenced the role of MTs as they are the major shareholder in sugar transportation from leaf (source) to sink (root, stem and seed) via phloem loading and nutrient transportation from roots to upper plant parts through xylem loading [2,3]. Thus, the efficient utilization of MTs not only could help researchers to increase grain yield directly but also expand arable land by improving tolerance mechanisms against various abiotic stresses, i.e., salt, drought, cold, heat and heavy metals (HMs) [4,5,6]. 

Efficient translocation of mineral elements and other substrates via MTs towards different upper plant parts contribute to the improvement of plant growth and development (PG&D). For example, Os*PTR6* MT is involved in the transportation of Gly-His and Gly-His-Gly peptides, showed substrate selectivity for dipeptides and tripeptides and its enhanced expression rate increased the growth of rice plants [7,8]. Similarly, *OsPRT7* (*OsNPF8.1*) is involved in long-distance translocation of methylated arsenic (As) to grains through its dimethyarsenate transporter activity [9]. For long-distance sugar transportation, almost every plant species contains at least one member of a sucrose transporter (SUC) family that first uptakes sugar molecules, mainly sucrose in the phloem, and then transport it to sink organs [10,11]. For example, in Arabidopsis, *AtSUC6*, a high affinity H^+^ symporter gene highly expressed in reproductive tissues translocated sucrose and maltose via plasma membrane (PM) to pollen tubes and synergid cells. In addition to pollen tubes, *AtSUC7* was also expressed in roots, and its expression is regulated in a sink-need-dependent manner [11]. In order to continue steady PG&D processes, a necessary supply of photosynthesis assimilates was required. In rice, upregulation of *OsHAK1* plays an important role in both vegetative growth and reproductive success in terms of root growth, pollen viability and fertility via its sucrose phosphate synthase activity [12]. Moreover, enhanced uptake of K^+^ renders plants less vulnerable to multiple abiotic stresses induced by osmotic stress. For a detailed overview on the role of MTs in PG&D, refer to Section 2. 

Both sugar and ionic homeostases in the cells, tissues and organs are regulated by a highly sophisticated membrane trafficking system. However, the source-to-sink relationship can be disturbed mainly due to the excessive uptake of salt and HMs and other environmental factors such as drought, cold and heat. Uptake of hazardous elements is often due to the poor selectability of MT/s as the chemical properties of essential minerals often matched with toxic elements, as argued in [13]. The excess amount of sodium entering into plant cells often disturbed the Na^+^/K^+^ ratio in the cytosol. In response, plants activate its Na^+^/K^+^ antiporter genes in order to achieve homeostasis. This tolerance mechanism is the same both in glycophytes and halophytes, but glycophytes are more susceptible as discussed in [14]. However, in rice, to tackle salt and drought stress, two members of the “Sugar Will Eventually be Exported Transporters (SWEETs)” family such as *OsSWEET 13* and *OsSWEET 15* are activated, and they enhanced the sugar supply across leaf and root organs in an abscisic acid (ABA)-dependent manner [15]. Overexpression of *TaZnF* in *A. thaliana* significantly increased tolerance to heat and cold stress as well as promoted early flowering [16]. Similarly, there are numerous studies in many plant species that evidenced the pivotal role of MTs in uptake, sequestration, translocation and detoxification of toxic HMs including cadmium (Cd) [17], chromium (Cr) [18], lead (Pb) [19], As [20], cobalt (Co) and zinc (Zn) [21] (for more details, see Section 3).

We provided an overview of the roles of MTs in PG&D in terms of their role in the improvement of plant architecture, seed yield, sugar transport and ionic balance for optimal plant growth by minimizing the impacts of external harsh environments. We also discussed the involvement of MTs in the uptake of toxic elements and explored the diverse and genotypic translocation capabilities in root and shoot organs. Lastly, we have tried to provide the recent understandings on how MTs respond to changing environments, for instance, strengthening of leaf sheath and increased sugar supply.

## 2. Role of Membrane Transporters in Plants Growth and Development

The biological role of proteins coding MTs in PG&D has recently been studied in various crop species including rice, maize, wheat and oilseed rape, etc., and has also been genetically verified in plants and in the model plant Arabidopsis. 

### 2.1. Plant Architecture, Seed Yield and Quality

Membrane transporters are involved in PG&D improvement by regulating plant architecture and nitrogen use efficiency (NUE) by enhancing tiller numbers and pattern and plant height. Moreover, they are also involved in the remobilization of mineral elements to the grains that all-together improve grain yield and quality. For example, in rice, *OsNPF7.2* is a nitrate transporter gene that is responsible in regulating the tiller number and grain yield [22]. Their results showed that the tiller number and seed yields were significantly higher compared with mutant lines in the overexpressed lines. Moreover, nitrate application was found to enhance the activity of *OsNPF7.2* in terms of faster growth of tiller buds, improved root length and number and fresh and dry weight compared to mutant lines. Furthermore, they also noted that in the transcript levels of genes responsible for the cytokinin pathway and cell cycle were enhanced in overexpressed lines in tiller buds. In another case, *OsNPF7.3*, a vacuolar membrane (VM) localized peptide transporter gene, responds to organic nitrogen (N) supplies and mediated N use efficiency that resulted in an increased number and filled panicles per plant, N content in grain and grain yield in rice as reported by Fang et al. [23]. Their results showed that the expression level of *OsNPF7.3* was higher in the vascular bundle of lateral roots and stem and regulated by organic N sources. In reverse, a reduction in plant growth, accumulation of amino acids in leaf sheaths and deterioration of leaf blades were recorded in the RNAi lines. However, in overexpressed lines at later reproductive stages, the N concentration was decreased in leaf but translocated towards grains. Thus, *OsNPF7.3* is involved the allocation of N and then improves the overall PG&D. In rice grain, the concentration of mineral elements is higher compared with other staple crops such as maize and wheat. The movement of mineral elements such as Zn and Cd to the grain is controlled by MT *OsZIP7,* a PM localized gene as investigated by Tan et al. [24]. Their results showed that the expression level of the above transporter gene was higher in parenchyma cells in both roots and nodes. Interestingly, in RNAi lines, the retention of Zn and Cd content was recorded in both the above organs, and Zn was distributed in entire root and away from leaves. Taken as a whole, *OsZIP7* plays a significant role in xylem loading and intra-vascular movement of Zn/Cd elements in nodes (for the thematic model, see Figure 1). 

### 2.2. Ionic Homeostasis, Detoxification and Transportation of Mineral Elements

In order to maintain steady PG&D, there is need to maintain the ionic balance of important minerals. For instance, *OsHMA3*, a tonoplast localized MT gene, maintained Zn homeostasis during growth periods in rice [30]. They reported that the root of rice accessions containing functional allele *OsHMA3* showed higher Zn tolerance compared to the lines with loss-of-function mutation *oshma3*. Moreover, in the lines with functional alleles, it also showed higher expressions of Zrt-/Irt-like protein (ZIP) family MTs such as *OsZIP4*, *OsZIP5*, *OsZIP8* and *OsZIP10* that are involved in the transportation and homeostasis of essential trace elements including Zn, iron (Fe) and manganese (Mn) [35]. These above findings suggested that *OsHMA3* plays a central role in the root of rice plants in maintaining cellular Zn content at the optimum level by its detoxification and storage into vacuoles. Although mineral elements are necessary for optimum PG&D, it can be toxic above a certain level. Thus, to adapt such hazardous condition, plants have evolved diverse Natural Resistance-Associated Macrophage Proteins (NRAMPS) through the process of alternative splicing (AS). For example, in rice, *OsNRAMP6* gene underwent AS and produced different NRAMP6 proteins such as full length (1-NRAMP6) and shortest (s-NRAMP6) [26]. These above variants are PM-localized proteins coding MTs responsible for the transportation of Fe and Mn. Moreover, in rice mutant *nramp6* plants, a significant deterioration of biomass was recoded that indicated its importance in PG&D. 

Essential mineral elements such as magnesium (Mg) are necessary for continuing optimum PG&D. To date, twelve Mg^+2^ transporters (MGTs) in total belong to a CorA-like gene family that has been identified in maize and are involved in the transportation and maintenance of Mg^+2^ homeostasis [36]. They reported that the *ZmMGT12* gene was very similar to a typical MGT in terms of two conserved TM domains and a GMN tri-peptide motif. The expression levels of *ZmMGT12* were higher in roots, stems and leaves but was more prominent in leaves. Interestingly, the higher expression rate in leaves was dependent on the rate of chlorophyll synthesis. Taken together, these findings suggested that *ZmMGT12* was responsible for the transportation of Mg^+2^ ion and probably for the chloroplast in the leaves. Similarly, in oilseed rape, a high affinity MT called *BnPHT1;4* belongs to a gene family PHT responsible in the acquisition of phosphorus (P) to facilitate seed germination and early seedling development [29]. Furthermore, results also showed that the expression level of *BnPHT1;4* was prominent in cotyledon at early seedling stages. In overexpressed plants, a higher rate of seed germination and seedling growth was recorded compared to wild type (WT). Moreover, changes in the accumulation of gibberellic acid (GA) and downregulation of abscisic acid (ABA) biosynthesis related genes were also observed in transgenic plants. Furthermore, they also showed that exogenous applications of GA and ABA were related to the increment of seed germination and recovery of advanced seed germination related phenotypes, respectively in the seeds of overexpressed plants compared to WT. These above-mentioned findings suggested that *BnPHT1;4* MT plays a role in the upregulation of GA and downregulation of ABA pathways related genes. Previous literature has also evidenced the antagonistic role of GA and ABA as primary endogenous regulators of the seed germination. For instance, GA promotes the transition of dormancy to germination, but the regulatory role of ABA is vice versa [37,38]. However, Pi deficiency resulted in the deterioration of total P contents in cotyledons. On the contrary, its exogenous supplementation increased total P contents in shoots and roots of overexpressed plants than compared to WT. These findings suggested that *BnPHT1;4* is involved in the acquisition and mobilization of P elements with respect to enhancing seed germination and seedling development at early stages in oilseed rape [29]. In another study on *B. napus*, it has been found that a total of 49 *PHT1* gene family members are involved in P acquisition and transportation. Among them, 27 *PHT1* gene copies are located in A-subgenome and 22 in the C-subgenome, and most of them are localized in the PM. Their results showed that a variable expression pattern was recorded in response to P deficiency. Moreover, these above *PHT1* gene copies were expressed higher under various mineral elements such as N, potassium (K), Fe and sulfur (S) and environmental stresses including salt and drought and phytohormones, for instance, auxin and cytokinin. In a nutshell, *PHT1* regulates nutrient homeostasis and responds to various environmental stresses in combination with their own family members as well as with other genes in a sophisticated network [39]. For the thematic model about ion homeostasis, detoxification and transportation of mineral elements, refer to Figure 1.

### 2.3. Remobilization of Photosynthates

In full daylight, plants produce plenty of carbohydrates including sucrose through the process of photosynthesis that, for time being, is stored in vacuoles. However, during the night, in order to continue normal PG&D, plants required its remobilization. For example, in maize, *SUCROSE TRANSPORTER2* (*ZmSUT2*), a sucrose /H^+^ symporter gene localized on the VM, is responsible for the export of stored sucrose to other parts, as reported previously [40]. Their results depicted that, in field conditions, the reduction in plant growth, deterioration in the lengths of tassels and ears and a smaller number of kernels were recorded in mutant (*zmsut2*) plants compared to wild type (WT). Moreover, in *zmsut2* mutants, two-fold accumulations of various sugars including sucrose, glucose and fructose were observed in leaves than compared with WT. These findings suggested that *ZmSUT2* is responsible for the remobilization of sucrose out from vacuoles and for transportation towards growing tissues that results in enhanced biomass and kernel yield. 

Recently, researchers discovered a novel MT family known as “SWEET” that majorly mediated sugar translocation across cell-membranes. The SWEET family is unique in its structure as it contains seven transmembrane domains possessing two internal MtN3 motifs (possibly originated from prokaryotic gene duplication event) that are involved in sugar transportation [41]. In order to study the role of the above MT family in sugar compartmentalization, Wang et al. (2018) isolated the *CsSWEET16* transporter gene from *Camellia sinensis* and then cloned and transferred it into Arabidopsis [42]. Sub-cellular localization analysis showed that *CsSWEET16* was located at the VM. Moreover, their results stated that the above transporter gene is involved in sugar transport across VM, and it responds to cold-stress in transgenic Arabidopsis compared to WT. In another study, *DsSWEET12* MT, a member of SWEET family, was identified in *Dianthus spiculifolius* and detailed its diverse role in PG&D, abiotic and biotic stresses [43]. Their qPCR data revealed that fold-change levels of *DsSWEET12* was enhanced in response to sucrose depletion, mannitol and H_2_O_2_. Furthermore, co-localization analysis stated that it was located in the PM. In Arabidopsis, overexpressed *DsSWEET12* plants showed greater tolerance to osmotic and oxidative stress and had longer root length and increased fresh biomass compared to WT, which mainly were dependent on continuous sucrose supplies. Modification of “SWEET” members at transcriptional level can further enhance their diverse roles in plants. For example, in rice, DNA Binding with One Finger 11 (OsDOF11) transcription factor (TF) mediated sugar translocation mechanism by promoting SWEET genes and also other sugar transporters [44]. In detail, T-DNA insertion mutant analysis showed that *OsDOF11* was expressed higher in vascular cells of photosynthesis apparatus and other sink tissues such as root, young seedling and leaf. In mutant *osdof11* lines, plants were semi-dwarf and had fewer tiller and smaller panicles compared to WT. On the other hand, in WT plants, the root length of young seedlings was significantly larger than mutant plants, which might be due to reduced sugar supplies in mutant plants. Moreover, chromatin-immunoprecipitation analysis showed that OsDOF11 TF was directly attached with the promoter regions of other sugar transporter genes such as *OsSWEET11/14* and *OsSUT1* (their expression levels are varied in different tissues), which suggested that *OsDOF11* regulates the expression levels of these genes in a coordinated manner to mediate the sugar transport system through apo-plastic loading. 

Source-to-sink translocation of sugar molecules (i.e., sucrose, raffinose and polyols) mainly determined PG&D and depends on their sophisticated controlled transportation to all organs via phloem loading [10,45]. Disturbance in phloem channels due to environmental cues can alter sugar homeostasis in different plant tissues [2]. In Arabidopsis, in order to regulate source-sink relationships, two sugar MT families such as SUC and SWEET play a specific role [46]. For example, in leaves, *AtSUC2* and *AtSWEET11/12* are known to play a role in phloem loading and were more expressed when sucrose translocation was either increased or decreased due to osmotic stress. However, in roots, the expression level of *AtSUC1* was upregulated, which indicated that it might be involved in sucrose unloading and root growth. On the other hand, SWEET transporters such as *AtSWEET12/13/15* was expressed higher in all tissue, but *AtSWEET14* was expressed differentially in root, stem and siliques, and *AtSWEET9/10* was expressed only in stem and siliques. The above findings suggested on-demand roles of these MTs in a source-sink relationship. To sum up, in normal conditions, the root always remained as a main sink, but carbohydrate allocation decreased to rosettes and roots under stressful environments. For the thematic model with respect to sugar transportation, refer to Figure 2. Table 1 describes the detail of the role of MTs in PG&D. For NCBI IDs of genes in Table 1, refer to Appendix A.

## 3. Membrane Transporters and Abiotic Stresses

### 3.1. Salt Stress

Salt stress is an ongoing and a potential risk in the future based on current and future research projects that show the potential number of salt tolerant patents, which has reached nearly up to 18,000 [53]. A global challenge for agriculture farming is multi-component stress regulated by multiple genes and genetic networks [14]. High salinity induces sodium ionic (Na^+^) stress in plants due to higher accumulations of Na^+^ in the cytosol that are prominently translocated towards upper part of plants. Na^+^ entering into the plant’s body certainly disturbs potassium ion (K^+^) homeostasis, which plays pivotal role in plenty of metabolic processes, especially in balancing cytosolic Na^+^/K^+^ as reviewed earlier [14]. Disturbances in Na^+^/K^+^ ratio due to excess salt stress resulted in the deterioration of several physio-biochemical and developmental pathways [54]. Studies on MTs involved in balancing Na^+^/K^+^ ratios have been evidenced in several plant species [14,54,55,56,57,58]. A possible strategy can be used to tackle ionic influx: compartmentalize it in vacuoles first, transport excessive Na^+^ from the affected cells and then distribute it into roots and other areal organs. For example, in Arabidopsis and halophytes, the above sophisticated process is performed by one of the MT families called high-affinity K^+^ transporters (HKTs) that are involved in long-distance transportation [54]. Another strategy can also be adopted by plants to handle Na^+^ toxicity in the cell by maintaining K^+^ homeostasis as reviewed in [14,55]. Whole genome sequencing technologies have explored where a large number of K^+^ transporters of HAK/KUP/KT family that belong to an amino acid polyamine-organocation (APC) super family spread across plant genomes. They play a central role in K^+^ uptake, and its translocation resulted in tolerance to salt induced osmotic stress and was also involved in the regulation of root morphology and shoot phenotyping. Moreover, in response to exogenous K^+^ supplementation, so far six TFs have been reported in regulating the gene coding MTs of the HAK/KUP/KT family. Furthermore, in order to activate their proper functioning (in uptake of K^+^), proteins of these MTs are phosphorylated by complexes of calcineurin B-like (CBL)-interacting protein kinases (CIPKs). These findings indicated how MTs are precisely responsible for K^+^ uptake, and their proper utilization in the entire plant body is regulated by several layers of controllers in a network [55]. 

Variation in salt stress tolerance mechanism is largely dependent on the genotypic diversity within a species or genera. Ideally, to obtain insights on the genetic makeup of different genotypes (to explore genetic diversity) of a plant species responsive to high salt stress, there is need to conduct thorough investigations under different salt concentrations. For instance, in rice, three genetically variable cultivars such as Koshihikari, Reiziq and Doongara (sensitive, moderately tolerant and tolerant, respectively) were investigated for their capability in maintaining ionic homeostasis in the cytosol and in being exported and transported from roots towards shoots [59]. Their results based on electro-physiological and qPCR data showed that two-times higher capacity of Na^+^ efflux was recorded in a tolerant cultivar in the elongation zone of a root organ. Moreover, Na^+^ efflux is partially controlled by the PM-localized Na^+^/H^+^ antiporter salt overly sensitive 1 (OsSOS1) MT, indicating that there are some other hidden mechanisms involved. In addition to the exclusion of Na^+^ from the root zone, there is another important hallmark involved in salinity stress tolerance, for example, the retention capability of K^+^ in the root zone. Behind the K^+^ holding, there are three complementary mechanisms such as the activation of the H^+^-ATPase pump (at both transcriptional and functional levels), increased tolerance of K^+^ efflux channels against negative impact of reactive oxygen species (ROS) and lastly higher-up regulation of *OsAKT1* gene copies compared with *OsGORK* in tolerant genotypes. These genotypes, which showed an increase in the above mechanisms (trait), can be incorporated in the salt stress breeding program for the development of commercial cultivars [59]. Similarly, in maize, Jiang et al. [60] explored natural sequence variations in HKT using a population of 54 maize varieties collected from diverse regions in China. Their results showed that the expression level of the *ZmHKT1;5* gene was significantly involved in tolerant inbred lines under salt stress environment, which resulted in balancing the Na^+^/K^+^ ratio and improved plant growth. Moreover, two single nucleotide polymorphic markers, for instance A134G and A511G, were present in the coding region in the above-mentioned MT, which was significantly associated with salt stress phenotypic traits at different salt concentrations. Furthermore, they produced transgenic tobacco lines by using favorable alleles of *ZmHKT1;5* from tolerant genotypes and displayed overexpressed lines that were more tolerant to salt stress than WT. Taken as a whole, the above two SNPs might produce new amino acids, resulting in higher expressions of *ZmHKT1;5* that may trigger the salinity stress mechanism by creating Na^+^/K^+^ homeostasis in the cytosol and enhance the activity of antioxidants in order to minimize ROS at low level.

Studies focusing on the overexpression of MTs shed a light on enhancing the protective shield in plants against high salt stress [54,55,59]. For example, in Arabidopsis, FERROCHELATASE 1 (FC1) is observed, which is a terminal enzyme of heme biosynthesis that is highly responsive to salt stress and involved in several metabolic and physiological processes [56]. They reported that AtFC1 plays a significant role in improving the Na^+^/K^+^ ratio by deterioration of Na^+^ concentration and improving K^+^ accumulation and protecting cell membrane lysis by upregulation of SOS1, which encodes a PM localized Na^+^/H^+^ antiporter gene. Interestingly, in overexpressed *AtFC1* plants, the expression levels of several high salt stress induced genes including *NHX1* and *AVP1* were reduced. This finding indicated that in order to regulate lower Na^+^ stress, AtFC1 might follow other routes rather than the mechanism adopted for Na^+^ sequestration. Similarly, in rice, Jadamba et al. [57] reported that overexpression of *OsEXPA7*, a key regulator of the cell wall, creates a protective shield against salt stress-induced oxidative stress. Their phenotypic data of overexpressed lines showed that *OsEXPA7* significantly enhanced salt tolerance, and expression was more prominent in the shoot apical meristem (SAM), root and leaf sheath. Furthermore, overexpression of *OsEXPA7* statistically enhanced K^+^ accumulation and reduced Na^+^ concentration in roots and leaves. Moreover, they also recorded that antioxidant activities were enhanced, and this resulted in the deterioration of ROS contents in overexpressed plants compared to WT. However, some stress-responsive proteins are highly conserved in both plants and animals that play a significant role in tackling various posed environmental stresses. We have such an example in wheat [61], where a *TaSAP17-D* gene, a member of a multigene family so-called stress associated protein (SAP), contains a conserved AN1/AN1 domain. Subcellular localization analysis showed that *TaSAP17-D* was localized in the cytoplasm, nucleus and cell membrane. Moreover, gene expression levels of *TaSAP17-D* were higher in response to a wide range of stresses including salt, cold, polyethylene glycol (PEG) and exogenous ABA. Moreover, in transgenic lines of Arabidopsis upregulation of *TaSAP17-D*, it was observed that when plants were subjected to salt stress, its role in salt stress alleviation was further confirmed [61]. 

Observing the structural variations of *cis-regulatory* elements of an MT can also open new avenues regarding the enhancement of salt stress response. For example, in halophyte *Aeluropus lagopoides,* the *HKT subfamily* II *(AlHKT2)* gene acts as a co-transporter of Na^+^ and K^+^ and helps in avoiding salinity stress [62]. Detailed investigations showed that full-length promoter D1 of the above MT has several cis-regulatory elements including MYB, W-box, MBS and ABRE, etc., and are involved in salt and other stress responses. Moreover, a 760 bp of D1 promoter region was isolated and cloned. An analysis of the GUS assay of transgenic plants (T2) confirmed the involvement of the promoter region of *AlHKT2;1 MT* in salt stress response. Hence, more studies need to be focused in the direction of molecular and functional characterization of K^+^ and Na^+^ MT systems.

### 3.2. Drought 

Dynamic climate changes induced severe drought stress and caused a decline in the productivity of crops in terms of seed yield and quality [63]. Thus, the main objective of agricultural biotechnology program is to develop high yielding plants with better survival abilities during water scarcity [64]. Dry and wet lab approaches alone or in combination for exploring new and well-utilized techniques already reported that MTs can bring solutions on the table to deal with drought. For example, in wheat, ElBasyoni et al. [65] conducted a comprehensive study on the role of cell membrane stability (CMS) in response to heat and PEG 600 stress. Results based on association mapping of single nucleotide polymorphisms (SNPs) were called from a panel of 2111 diverse spring accessions with a CMS trait. Interestingly, several significant SNPs were found to be highly associated with CMS. Moreover, candidate genes in the QTL regions were mainly related to solute transporters of CMS and other biochemicals responsive to abiotic stress. In the field trial data, they found that selected accessions (based on significant SNPs screening) produced more seed yield under the above abiotic stress environment. In a nutshell, a genome-wide design breeding program of CMS can be used for the selection of parental genotypes for producing commercial cultivars that can grow well in drought-affected soils. Comparative studies on close relatives can be utilized for exploring crop stress tolerance behavior. For example, Hasan et al. [66] investigated the response of maize and sorghum plants under drought (10 d withholding water) in field conditions. Their results showed that drought stress demonstrated severe impact on maize than compared to sorghum in terms of physiological attributes such as shoot fresh, dry weight and leaf water content; gas exchange capacity; and water use efficiency (WUE). Moreover, gene expression data regarding four PM intrinsic proteins (PIPs) of the aquaporin gene family, such as PIP1;5, PIP1;6, PIP2;3 and TIP1;2, showed that PIP1;5 and PIP2;3 in leaves and roots, respectively, were highly regulated in response to drought stress in sorghum but were not expressed in maize. The above two proteins possibly play a central role in water transport for tackling drought severity. Furthermore, their gene expression data suggested that in both species *PIP1;6* probably contributed to CO_2_ transportation and *TIP1;2* in water transport under control conditions but not during drought stress. 

Cloning of well-known MTs from tolerant plants and then genetic transformation to less tolerant plants can enhance coping capacities of drought stress. For example, Zheng et al. [67] recently cloned PIP2;3 from halophytes *Canavalia rosea* and then transformed in yeast and Arabidopsis to evaluate its heterologous expression systems under drought induced high osmotic stress. The expression level of *CrPIP2;3* was enhanced under osmotic stress. Its overexpression further relieves the drought-induced impact by improving water homeostasis but not ROS scavenging in transgenic Arabidopsis plants. In plants, HAK MTs helps plants to cope with drought stress by improving K^+^ acquisition and transportation. The question is as follows: how it is regulated in response to the above stress? In order to answer this question, Chen et al. [68] investigated the regulatory role of *OsHAK1* in rice under osmotic and drought stress and found that its expression level was transiently increased in water deficit shoots and roots. On the other hand, knockout mutants *oshak1* plants displayed a reduction in growth at both vegetative and reproductive stages under drought stress environment. However, overexpressed plants showed better tolerance to drought stress in terms of reduced lipid peroxidation content, higher antioxidative activities (POX and CAT), positive regulation appearing in rice channel genes (*OsTPKb and OsAKT1* involved in K^+^ homeostasis and stress response) and recovery at the reproductive stage that enhanced 35% grain yield compared with WT. In a nutshell, *OsHAK1* acts directly and is a positive regulator of downstream genes with respect to better coping with the drought stress trait. Thus, this gene can be incorporated in molecular breeding programs related to drought stress tolerance.

Furthermore, MTs also responds to stress through the regulation of hormonal levels in the tissue where hormones are produced (hormonal homeostasis), and then they are transported to the affected cells via respective intra-cellular signalling pathways. ABA predominantly produced in vascular tissues and exerts its hormonal activity in various cells through the ABA signalling pathway under developmental and environmental cues. For example, in *A. thaliana*, Née et al. (2017) screened an ABC transporter gene, AtABCG25, based on its sensitivity to ABA and reported that it was mainly expressed in vascular tissues and localized in the PM [38]. Their results further showed that in insect cells, AtABCG25 plays a role in ATP-dependent ABA transportation. Moreover, in overexpressed plants, higher leaf temperatures and impacts on stomatal regulation were observed. In another case [69], in yeast, AtABCG40 (also called a pleotropic drug resistance transporter 12, AtPDR12) that is also localized in the PM plays a role in ABA transportation. Comparative overexpressed *AtABCG40* and mutant *atabcg40* results showed that ABA uptake increased in the first case (yeast and BY2 cells) and decreased in the former case (protoplast). Moreover, ABA responsive genes and stomatal closure in mutant plants were strongly delayed compared to overexpressed ones under exogenous applications of ABA, which indicated reduced drought tolerance in the loss-of-function atabcg40 mutant plants. These results suggested the coordination between the process of ABA-dependent signalling and its transportation to the required cells as guard cells in this case. Similarly, AtABCG22 was expressed higher in guard cells and is required for the regulation of stomatal opening/closure under drought stress. In Arabidopsis mutant plants, increased leaf temperatures and transpiration rate/water loss were recorded compared to WT. They further discovered that, in double mutant plants, (1) atabcg22 enhanced the water loss of srk2e/ost1 (defective of ABA signalling in guard cells) mutant plants and (2) altered the phenotype of nced3 (defective in ABA biosynthesis) mutant plants compared with WT. Thus, AtABCG22 plays a dual additive function both in ABA signalling and biosynthesis [70]. In peanuts, *ABA transporter-like 1* (*AhATL1*), a member of the ABCG transporter family, is localized in the PM and is expressed higher in water-deficient and ABA-treated conditions [71]. In a recent study [72], they showed that *AhATL1* expression levels and its distributions were enhanced rapidly in a second round of drought stress compared with the first round. Similarly, under recovery conditions (normal watering), the expression of *AhATL1* and ABA contents was also higher in the second round than the first round. These findings suggested that AhATL1 is a memory-type gene that respond first to drought stress and later upregulates endogenous ABA contents. Overall, the above findings open new windows for genetic engineering as well as designing the fast-forward drought tolerance breeding program. 

### 3.3. Heat Stress

In order to achieve the goal of “Zero Hunger” as proposed by FAO, heat stress is one of the major obstacles as it drastically disturbs the plant’s physio-biochemical processes, gene expression regulation networking and metabolome and proteome levels, as reviewed in [73]. As plants are sessile, in order to withstand dynamic confrontational environments, they have evolved significant alterations in their adaptive behaviour. To date, a variety of genes coding MTs has been reported, which plays a pivotal role in diverse biological systems including plant metabolism and its production resulting in the increase in heat tolerance mechanisms in plants [74,75]. Regulatory proteins of MTs play a significant role in the regulation of PG&D and in the enhancement of coping potentials against a faced threat. For example, in plants, the zinc finger (ZnF) protein belongs to a C3HC4-type TF family that plays an important role in multiple developmental and environmental cues. In wheat, a gene expression analysis of *TaZnF* showed that its expression starts at the stage of post-anthesis period 3-5DAA (a sensitive stage related to yield) and was expressed higher in seeds [16]. In transgenic Arabidopsis, the overexpression of *TaZnF* resulted in increased tolerance to both basal and high temperature stresses compared to WT and also improved PG&D. Moreover, overexpressed lines showed earliness in flowering, larger primary roots, more lateral branching, increased leaves number and size, enhanced fresh biomass and overall improvement in PG&D and ultimately yield. While plants are continuously facing thermal stress, maintaining membrane integrity at an optimum level is an important aspect of the respective tolerance mechanism that should be thoroughly investigated. For example, in Arabidopsis, P4-type ATPases play an important role in the stability of membrane systems as they take part in transportation and homeostasis of phospholipids. In total, twelve P4-type ATPases have been reported in Arabidopsis so far. Among them, aminophospholipid ATPase6 (ALA6) is one of the members of the above family (P4-type ATPase). Researchers reported that the loss of function mutation (*ala6*) resulted in sensitivity to both low-level and high-level temperature treatments. On the other hand, in the overexpressed ALA6 lines, resistance to heat stress at the seedling stage was recorded. Moreover, they also discovered that in point mutation (ALA6, one bp at conserved functional site), heat susceptibility of transgenic plants was observed similarly to the loss-of-function *ala6* mutant. Moreover, in *ala6* mutant plants, higher ion-leakage was observed, which indicated that lipid flipase activity is regulated by ALA6 in heat stress tolerance. Thus, ALA6 under heat stress plays a vital role in membrane stability [76]. 

In response to heat stress, several different lipids accumulate in various cell organelles as an adaptive strategy. For example, in Arabidopsis, increases in galactolipids (containing linoleate, 18:2) in chloroplasts; phospholipids (containing palmitate, 16:0; stearate, 18:0; and oleate, 18:1) in the endoplasmic reticulum and PM; and triacylglycerol (containing α-linolenate, 18:3) and hexadecatrienoic acid (16:3) as lipid droplets were recorded in leaves in response to heat stress, as reviewed in [77]. Response of cellular organelles at the global network level can open new avenues, contributing to the overall heat stress mechanisms in plants. As we know, chloroplast is a central player for steady PG&D and its survival; moreover, the performance of chloroplast is closely associated with the cell’s general status when facing heat stress. In order to explain the vital role of chloroplasts, Paul et al. [78] examined chloroplast morphology and proteome composition under single and repetitive heat stress periods in Arabidopsis that were prolonged until two weeks. They depicted that in a single heat stress condition, a significant alteration was observed compared to repetitive ones where adaptive behavior was noticed. Moreover, changes in chloroplast morphology in terms of adjustments and adaptation were dependent on protein translocation efficiency, as evidenced in mutant plants of two chloroplast translocon units such as TOC64 and TOC33. Interestingly, in the adaptation period to repetitive heat stress, the *toc33* mutant displayed an accumulation of HSP70 protein, while *toc66* showed higher protein contents responsible for the determination of thylakoid membrane structure compared to WT plants. At reproductive stages, rice plants respond variably to heat stress in terms of panicle initiation, flowering time and grain filling. Thus, genotypic screening of rice accessions at grain filling stage can help assist the development of future breeding plans in order to cope with yield loss due to heat stress [79]. 

In addition to lipid accumulation as mentioned above, the integrity and constant PM fluidity are also very important for proper Ca^+2^ movement across PM and the consequent induction of multiple heat shock proteins (HSPs) in response to heat stress. Previous research on fatty acid showed that the disparity between saturated and unsaturated fat levels was found to be increased and caused a negative influence on the membrane’s integrity under heat stress. Additionally, at elevated temperatures, the content of polyunsaturated fatty acids rises [80,81]. Temperature is the most important contributing factor in regulating the degree of unsaturated fatty acid chains; therefore, it has a significant impact on the properties of diaphragms [82,83]. The numbers and locations of double bonds within fatty acid chains are modified by desaturation through oxygen-based reactions [84]. In plants, the family of fatty acid desaturase (FAD), which plays a significant role in adapting to high-temperature stress, has been studied. Among the eight Arabidopsis FAD family members (FAD1 to FAD8), FAD2 and FAD3 were found on the ER membrane, while others include plasmid dictatorships [85]. Evidence from studies on model plants indicated that Ca^+2^ movement occurred within seconds after thermal treatment through their respective permeable channels that are present in the PM. Detailed investigation on the *A. thaliana* genome showed that >40 putative Ca^+2^ channels are located and might be considered as candidate heat sensors [86]. Plants used a Ca^2+^ sense and signalling system as a “systemic acquired acclimatation” strategy to adapt in response to heat stress. For example, in *A. thaliana*, CNGC6 (cyclic nucleotide-gated ion channel 6) is involved in the regulation of heat-inducive Ca^2+^ influx and the activation of expression of several *HSPs* during the plant adaption period under heat stress [87]. Moreover, transportation of free Ca^+2^ by binding CaMs (calmodulins) to the PM enhanced the signalling cascades, which might trigger the activation of various heat shock factors and HSPs and their interconnected sophisticated networks [88]. A study on the moss plant [89] suggested that the intensity of heat shoch response (HSR) modulated after a minute of thermal stress is regulated by Ca^2+^ transportation across PM. Their results showed that Ca^2+^ permeable channel in the PM is transiently activated even by mild changes in the temperature or chemistry of membrane fluidity. Taken together, Ca^2+^ channels in the PM regulate HSR response and play a role in the establishment of temperature tolerance in the plant body. 

### 3.4. Cold Stress

Low temperature is a limiting factor for PG&D and productivity as well as geographical distribution of many crop species. Thus, plants must be capable in responding immediately to temperature changes in order to adapt well to the growing environment and to avoid metabolic losses. Plants have a variety of MTs that play a significant role for the continuity of vital cellular processes, for instance, ionic homeostasis, maintenance of osmotic pressure, signal transduction and sequestration under cold stress condition [90]. For example, PIPs play a role in water uptake under both normal and stress conditions. An integrated approach based on physiological, cell biology and transcriptional profiling attributes stated that PIPs are involved in cold stress response [91]. Their results suggested that the expression levels of two PIPs such as *PIP1;4* and *PIP2;5* were upregulated, and the protein of *PIP2;5* and total PIP contents also increased in order to adapt under cold stress. These findings indicated that *PIP2;5* plays a pivotal role in tackling the impact of cold stress. Plants interaction with the surrounding environment is a result of cuticle layer (composed of wax and cutin), which is made of lipids and covers most of its areal surfaces. Thus, the protection of the cuticle layer to external harsh environment can be an important adaptive strategy of plants. For this purpose, plants have ABC subfamily G members (ABCG) that are involved in the transportation of lipid molecules of the cuticle to the plant’s upper surface or outside. For example, *Thellungiella salsugineum* is an extremly stress-tolerant plant that contains thick cuticle layers [92]. Cloning of *TsABCG11* from the above plant and transformation in its close relative Arabidopsis demonstrated a lower rate of water loss, and chlorophyll content was recorded in overexpressed lines compared to WT under 4 °C. In the same transgenic lines, significant changes were observed in wax components (C31 and C33 alkanes) and cutin (C18:2), resulting in the improvement of wax and cutin contents in the cuticle layer.

Moreover, with the transgenic technique, there is need to find novel genes and protein using whole genome transcriptomics and proteomic approaches. Comparative transcriptome profiling can provide millions of treatment-dependent reads that can be aligned with reference genomes (if available) and de novo (if not) and ultimately will provide thousands of novel transcripts/genes. For example, in rice, mRNA sequencing of two contrasting cultivars, i.e., Y12–4 (cold-tolerant) and 253 (cold-sensitive) showed that a total of 42.44–68.71 million reads were obtained and later on aligned to nearly 30 and 29 thousand genes, respectively [93]. In the results, a majority of cold tolerant DEGs were found in tolerant genotype. Among all, they explored one cold responsive gene, which is low temperature growth 5 (LTG5) that coded for UDP-glucosyltransferase. This enzyme belongs to a super family of membrane-bound enzymes that are involved in the glycosylation of endoplasmic reticulum (ER)-localized additional glycoproteins. Here, it acts as an MT and transfers sugar molecules across EM membranes. It also play a pivotal role in regulating the various activities of flavonoids including stability, availability and biological activities [94,95]. Its upregulation under cold stress indicated that the *LTG5* gene plays a role in the cold-stress mechanism in rice. Moreover, UDP-glucosyltransferase is located in a GSA1 QTL region that is responsible for grain size and abiotic stress response, as recently reported in [96]. They stated that grain size is controlled by cell proliferation and expansion and regulated by glucosyltransferase activity towards flavonoid mediated auxin homeostasis and other related genes. However, in abiotic responses, GSA1 remobilizes metabolic flux from lignin biosynthesis towards flavonoid biosynthesis, resulting in the accumulation of glycosides which prevent rice plants from abiotic stress. 

There are very few literatures available that can answer the very basic question about the earliest response of plants to a drop in temperature. Plant organelles, for instance, PM and its nearby extracellular and cytoplasmic cites are so-called primary checkpoints for sensing temperature fluctuations and the following events include signal transduction and remobilization of solutes by MTs. To obtain insights on cold sense and response mechanisms, Kamal et al. [97] investigated short-term changes in PM proteome using a phosphor-proteomic method powered by a mass spectrophotometry approach in Arabidopsis exposure to cold stress for 5–60 min. Their results revealed that rapid changes occurred in proteins responsible for ionic homeostasis, transportation of solutes and other proteins, protein modification, cytoskeleton organization, vesical trafficking and signal transduction processes. Moreover, motif phosphorylation and protein kinases (PKs)-substrate network analysis suggested that plenty of PKs such as MAPKs, CDPKs, RLKs and their substrates might also be played a role in cold response mechanisms. In another study on PM proteome [98], Takahashi and his colleagues provided further insights on freezing tolerance by treating Arabidopsis suspension cultured cells with cold acclimation and ABA. With respect to the insights on changes in cellular responses to the above treatments, the PM proteome was analysed using a label-free protein quantification method. Data revealed that a total of 841 proteins were accumulated during the transition of the growth phase, cold acclimation and ABA environments. Among them, 392 proteins were accumulated in respond to the progression of growth phase and were divided to several functional groups, suggesting that PM physiology is dependent on the growth phase. Moreover, in response to cold acclimation, ABA plays a role in signal transduction, which resulted in the changes in PM proteome. As a whole, modifications in PM proteome in response to the transition of the growth phase impact PM proteome-related cold acclimation and ABA, which may influence its tolerance capability to freezing. 

Explorations of cold response at genotypic/species levels can further unveil the overall tolerance mechanism. For example, in maize [99], two widely used inbreds, i.e., B73 and Mo17 and 37 other inbreeds (grouped into stiff-stalk, non-stiff stalk and tropical clades) exhibited variable cold tolerance behaviour that was evaluated at seedling stages. Recombinant inbreed (B73 × Mo17) line (RIL) populations were developed (total of 97 RILs) to identify the QTLs responsible for the cold response at seedling stages. Based on association with three phenotypic traits such as leaf colour, chlorophyll content and tissue damage, two QTLs on chromosome 1 and 5 were detected. The DEGs in these two QTLs responsive to cold stress are the genes with similar putative functions such as auxin and GA as well as general abiotic stress response. Thus, breeding with these QTLs can protect maize plants from low temperate stress right after germination, and breeders are suggested to allow early sowing and to provide prolonged growing period, resulting in high crop yield. 

### 3.5. HMs Stress

With essential mineral elements, toxic HMs often coexist in arable soil, which seriously threatens global food productivity. Due to similar chemical properties of essential mineral elements and inessential HMs, some of the MTs such as Natural Resistance-associated Macrophage Proteins (NRAMPs) transport them to upper plant parts. In dwarf polish wheat (*T. polonicum*), TpNRAMP3, a member of the NRAMPs family, was identified for being responsive to PM-localized protein, and it is expressed higher in leaf blades and roots and first noted at jointing, booting and grain-filling stages, respectively [21]. In detail, *TpNRAMP3* was cloned and transformed into yeast and Arabidopsis. The results revealed that the expression level of *TpNRAMP3* increased in response to Cd and Co but not Zn in yeast; however, in Arabidopsis, expression was higher in response to Cd, Co and manganese (Mn) and but not for Zn and Fe in roots, shoot and the entire plant. Moreover, their data stated that the above MT did not disturb transportation of the above metals from roots to shoot. Thus, these results suggested that *TpNRAMP3* plays a role in the uptake of Cd, Co and Mn. After entering HMs, the plant body must be degraded and detoxified; otherwise, it can deteriorate PG&D and ultimately yield. In rice, stress membrane proteins (SMPs) globally respond to abiotic stress including HMs. Recently, Zheng et al. [100] reported that *OsSMP1*, a member of SMPs family is localized on the cell membrane. Data stated that overexpression of *OsSMP1* significantly enhanced tolerance mechanisms to multiple stresses such as salt, cold and HMs including Cd and copper (Cu). Overall, they suggested that *OsSMP1* positively regulated abiotic stresses, i.e., salt, cold and HMs, through the ABA-mediated pathway and potentially can be utilized for rice stress breeding.

Among MTs, ABC transporters play a pivotal role in the uptake and translocation of a wide-range of metabolites and xenobiotics, including HMs, and have been studied in several plants’ species including Arabidopsis [101,102], rice [103] and *B. napus* [17]. Firstly, the characterization of ABC transporters was performed in popular species, and later on their orthologs in other plants were transformed and studied [101]. For example, *PtABCC1*, a member of ABC transporters, was cloned from *Populus trichocarpa* and overexpressed in both *P. trichocarpa* and Arabidopsis [101]. Data revealed that transgenic plants were more tolerant to exogenously applied mercury (Hg) stress, and overexpressed plants accumulated 26–72% and 7–160% Hg in Arabidopsis and popular, respectively, compared to WT. In a similar study, Wang et al. [102] cloned *PtoABCG36*, another member of the ABC transporter family, from *P. tomentosa* and reported that it is expressed almost in all plant organs including root, stem and leaves and plays a significant role in Cd tolerance in Arabidopsis. In detail, its expression was increased just after 12-hour exposure to Cd stress. The results revealed that, in overexpressed lines, Cd accumulation was significantly decreased in terms of net Cd^2+^ efflux compared to WT. These findings suggested that *PtoABCG36* acts as a Cd extrusion pump contributing to Cd tolerance by decreasing its contents, indicating that transgenic plants with this MT can be a promising method for cultivating crop plants in Cd-affected soils. In order to broaden the scope of ABC transporters, in *B. napus*, Zhang et al. [17] reported that a total of 314 ABC transporters were identified from high-throughput sequencing data. All 314 MTs further categorized into eight subfamilies, including ABCA-G and ABCI. Further analysis suggested that their expansion was a result of allele duplication. Most ABCs (233/314) were also verified by comparing RNA-Seq data at seedling stages, and among them (233, BnaABCs), 132 were DEGs, and 84 were significantly responsive to Cd stress. Lastly, the analysis of *cis-regulatory* elements suggested that eight Cd-responsive DEGs showed significant variation, indicating their role in both environmental (abiotic stress and hormonal signaling) and developmental cues.

Similar to ABC transporters, HM ATpase (HMA) also plays a significant role in metal ion transportation across membranes. For example, in wheat, *TaHMA2* can transport Cd^2+^ and Zn^2+^ through cellular membranes [104]. Moreover, two motifs such as CCxxE and CPC in the N-MBD and N/C-terminal are mainly involved in the transportation of above metals. They produced four types of transgenic Arabidopsis plants: (1) overexpressed plant with normal *TaHMA2* functioning, (2) *TaHMA2* derivative (substitution of glutamic with alanine in CCxxE motif) plants and (3) plants containing truncated N/C-terminal and mutants with cysteine in the N-MBD region. The results stated that in first two types of plants, there was an increment in PG&D attributes such as root length and fresh biomass and also an enhancement in the transportation of Cd and Mn ions from the root to shoot. In third type, tolerance and translocation activities of *TaHMA2* was impaired. In last type of plants, tolerance and transportation activity of *TaHMA2* deteriorated compared to WT. These findings suggested that cysteine plays a significant role in binding Zn^2+^/Cd^2+^ and translocation from root to shoot. In another comparative study on rice, maize and sorghum, Zhiguo et al. [105] detailed the role of P_1B_ type HMA in metal ion transportation and PG&D. A total of 31 P_1B_ type HMA genes were identified, including eleven each in maize and sorghum and nine in rice. Sequence composition and phylogenetic analysis suggested that the above HM transporters were categorized in two sub-families, and four of them were duplicated in tandem. Members of HMA were diverse in terms of gene expression under Cd/Cu treatment and tissue specificity. The above findings suggested the utilization of HMAs (*TaHMA2 and* P_1B_ type HMA) in crop HM-stress breeding programs. 

An exploration of genotypic variation at physio-chemical, molecular and genomics levels in coping with HMs can, overall, broaden the scope of HM tolerance mechanisms [18,106,107,108,109,110]. In maize, a group of researchers investigated the impact of four metals including Cd, Cu, Zn and Ni on roots’ young leaves (14 days after sowing, DAS) and mature leaves (21 DAS). Data demonstrated that the concentration levels of all four metals varied in all organs but more prominently in roots. Interestingly, in older leaves, metal transportation caused the accumulation of ABA, resulting in the closure of stomata and ultimately a reduction in photosynthesis and fresh weight. Moreover, accumulations of three metabolites such as Tocopherols, polyphenols and flavonoids were enhanced in shoots in response to Zn, Ni and Cu stress. Similarly, activities of antioxidants, for instance, SOD and DHAR, were upregulated in the roots of Cu and Cd treated plants; however, APX was upregulated only in mature leaves. These findings suggested that the organ’s response to metal ions is also associated with the upregulation of antioxidants. In the last decade, a tremendous amount of work has been conducted in terms of sequencing many plant genomes, thereby opening new avenues to examine, e.g., PG&D and environmental cues. In *B. napus*, Gill et al. [18] explored whole genome transporters (transportome) under Cr stress using high-throughput de novo sequencing data. The results depicted that a total of 2867 and 2849 MTs were found in two contrasting cultivars, ZS 758 (Cr-tolerant) and Zheda 622 (Cr-susceptible). In detail, under Cr stress condition, 295 MTs showed upregulation in ZS 758, and 268 MTs were expressed higher in Zheda 622. Among these, three novel MTs, i.e., BnaA04g26560D, BnaA02g28130D and BnaA02g01980D were found and their functions were similar to water transport through cell membranes. Taken together, a well-coordinated organ, antioxidant and MT response is required for sequestration, translocation and detoxification with respect to metal ions in order to increase plant stress resistance mechanisms. Table 2 and Figure 3 describe the detailed roles and mechanisms of MTs in plants under abiotic stresses. For NCBI IDs of genes in Table 2, refer to Appendix A.

**Table 2 ijms-22-12792-t002:** Role of membrane transporters in abiotic stress tolerance.

Abiotic Stress	Transporter Protein	Plant Species	Tissue/Organ	Biological Role	Ref.
Salt	AlHKT2;1	*A. lagopoides*	Leaf/shoot /root	Na^+^/K^+^ co-transporter gene prevents plants from salinity stress	[62]
	OsCam1–1	*O. sativa*	Leaf	It is involved in signaling, hormone-mediated regulation, transcription, lipid, carbohydrate and secondary metabolism, photosynthesis, glycolysis, TCA and glyoxylate cycle under salt stress	[111]
	AtFC1	*A. thaliana*	Roots, cotyledon, root, shoot, leaf and flower	It enhances K^+^ accumulation and prevents cell membrane lysis; it also upregulates the expression levels of NHX1 and AVP1	[56]
	ATG8	*A. thaliana*	Root/cortex cells	It plays a role in nutrient remobilization following salt induced autophagy	[112]
	PRE1/AAP1	*A. thaliana*	Root	It enhanced uptake and transportation of proline and prevented proline degradation	[113]
	OsAKT1	*O. sativa*	Root/elongation zone and shoot	Retain K^+^ in root to balance Na^+^/K^+^ ratio	[59]
	ZmHKT1;5	*Z. mays*	Leaf	Balances Na^+^/K^+^ ratio and improves plant growth	[60]
Drought	CrPIP2;3	*A. thaliana*	Germinating seed, seedling and root	It plays pivotal roles in maintaining water and nutrition homeostasis	[67]
	PIP1;5/PIP2;3	*S. bicolor*	Root and leaf	Maintains WUE	[66]
	H^+^-ATPase	*C. sinensis*	Leaf	Maintenance of K^+^ homeostasis in mesophyll cells	[114]
	OsHAK1	*O. sativa*	Root and shoot	Involved in K acquisition, translocation and homeostasis by upregulating *OsTPKb* and *OsAKT1*	[68]
	OsNAC5/6/9/10	*O. sativa*	Root	Target genes were involved in transmembrane/transporter activity, carbohydrate metabolism, vesicle and plant hormones	[115]
Cold	HSP70-16/VDAC3	*A. thaliana*	Seed, endosperm and embryo	Activation of the opening of VDAC3 ion channels, ABA transportation from endosperm to embryo and then inhibits seed germination	[116]
	CsSWEET16	*A. thaliana*	Leaf and flower buds	Sugar transport across vacuoles and cold tolerance	[42]
	TsABCG11	*A. thaliana*	Root, stem leaf, rosette leaf, flower and silique	Thickening the leaf cuticle layer (wax and cutin) by exporting cuticle lipid molecules to prevent plants from cold stress	[92]
	AlTMP2	*A. littoralis*	Root and leaf	Improves membrane stability	[117]
	AtPIP1;4/AtPIP2;5	*A. thaliana*	Root and shoot	Plays a role in cold acclimation and freezing tolerance	[91]
	VAB3/NHX2/NHX5	*E. botschantzevii*	Shoot	Cold acclimation	[118]
	SOS1/VP2/HA3	*E. salsugineum*	Shoot	Cold acclimation	[118]
Heat	TaZnFP	*A. thaliana*	14-day seedling	Larger primary roots, more lateral branches, increased in leaf size and numbers, promotes early flowering and enhanced fresh biomass	[16]
	P4-type ATPase	*A. thaliana*	14-day seedling, rosette leaf, flower (stamen and pistil) and silique	It is involved in flipping lipids that cope with heat stress	[76]
	OsSUS	*O. sativa*	Flag leaf, stem-sheath and spikelet	It acts as a signalling molecule to mediate source and sink relationships under heat stress	[119]
HMs	TpNRAMP3	*T. polonicum, Polish wheat*	Leaf and root	Transport Cd, Co and Mn but does not transport Fe or Zn, which induced HM toxicity	[21]
	PtABCC1	*P. trichocarpa/A. thaliana*	Root	It enhances the accumulation and tolerance to Hg	[101]
	PtoABCG36	*A. thaliana*	Leaf/stem/root	It acts as an extrusion pump to decrease Cd uptake and enhance tolerance to Cd stress	[102]
	PtoABCG36	*O. sativa*	Root and shoot	Export Cd from root and enhance Cd tolerance	[103]
	OsSMP1	*O. sativa*	Leaf	Acts as a positive regulator of Cd and Cu tolerance via ABA-dependent pathway	[100]
	LmSAP	*N. tabacum*	Leaf and root	Enhanced accumulation of Cu, Cd and Mn, decreased H_2_O_2_ content, upregulated SOD, POD and CAT activities and stress related metallothioneins, i.e., Met1-5	[120]
	AtCNGC1/10/13/19	*A. thaliana*	Primary root and seedling	Plays a role Pb toxicity by reducing its uptake	[121]
	AtCNGC11/13/16/20	*A. thaliana*	Primary root and seedling	Plays a role Cd toxicity by reducing its uptake	[121]
	SaNramp6	*A. thaliana*	Root, stem and leaf	Improves Cd accumulation	[122]
	OsLCT1/OsHMA2/ OsZIP3	*O. sativa*	Root and shoot	Co-expression of HM transporters improved root and shoot lengths under Zn and Cd stress	[123]

**Figure 3 ijms-22-12792-f003:**
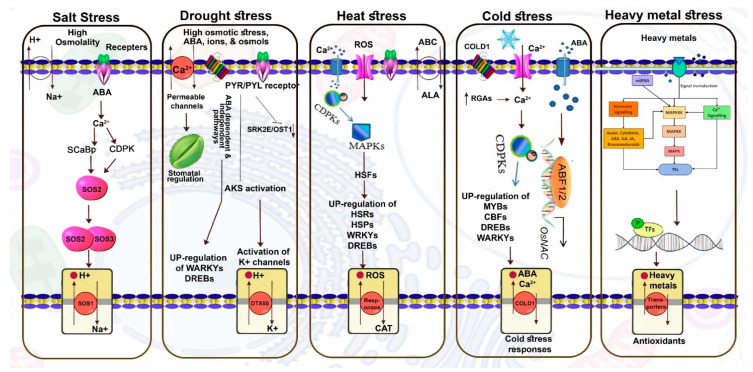
**Role of MTs in response to abiotic stresses**. Under salt stress, high osmolality and Na^+^/H^+^ imbalance detected by plant receptors, which activate ABA and Ca^2+^-driven pathways that further regulate the SOS system to maintain Na^+^/H^+^ balance in plants cells. Under drought stress, osmotic stress increases the amount of ABA, different ions and osmolytes in order to maintain osmotic balance and triggers various molecular and metabolic pathways including Ca^2+^, which regulates stomatal openings. PYR/PYL receptors activate the AKS system, which regulates K^+^/H^+^ ion flux across the cell membrane. Various ABA-dependent and ABA-independent pathways regulate the stress-related transcriptional factor (TF) pathways. Under heat stress, ABC transporters and overproduction of ROS activate defense mechanisms in the presence of Ca^2+^, and then both CDPKs and ROS regulate the MAPK system, which upregulates heat shock factors including HSRs, HSPs, WRKYs and DREBs. These pathways maintain ROS and antioxidant (CAT) balance. Under cold stress, COLD1, Ca^2+^ and ABA act as stress receptors and trigger CDPK-driven systems and other TFs systems. Under heavy metal stress, after signal transduction, various hormones, miRNA and Ca^2+^ activate the MAPKs system in order to regulate TFs, which maintain heavy metals/toxic agents and anti-oxidants. In addition to the text, for further details about the model, readers can also refer to [85,124,125,126,127,128,129].

## 4. Closing Remarks

The global role of MTs in the improvement of PG&D and abiotic stress tolerance has recently been appreciated extensively. As MT networking spreads in all plant parts, they efficiently regulate cellular and long-distance trafficking that usually occurs at the plant-stage and is environmentally driven. In PG&D, OsNPF7.2, OsNPF7.3 and OsZIP7 genes in rice improved plant architecture by enhancing root length, tiller numbers and grain yield as well as being involved in the translocation of N and other mineral elements such as Zn and Cd towards grain. For continuous normal functioning, plants require a regular supply of sugar molecules, including sucrose, fructose and glucose, etc., and water and mineral elements. For this job, plants have plenty of MTs (almost 18% of Arabidopsis genomes are associated with cellular membranes [130] and ~2850 putative unigenes in B. napus act as MTs [18]). They remobilize photosynthesis assimilation with the help of SUT, SUC and newly discovered SWEET genes families. Moreover, MTs are specialized in transporting sugar molecules through VM and PM from the leaf to growing tissues such as roots, rosette leaves and seeds. However, under stressed conditions (i.e., salt, drought, cold, heat and HMs), source-to-sink coordination is disturbed. Under unfavorable conditions, plants activated their stress-specialized MT genes families such as HKT, FC, EXPA and SAP in response to salt stress; PIP and ATPases reacted to drought stress; ZnF and P4-type ATPase reacted to heat stress; PIP and ABC reacted to cold; and NRAMP, SMP, ABC and HMA reacted to HMs stress/s. Mainly, these MTs maintained the Na^+^/K^+^ ratio by uptaking more K^+^ and controlling Na^+^ efflux at VM and PM; controlling the homeostasis of mineral elements in the cell; and they were exported to upper parts such as grain to enhance grain quality and water transmembrane activity under salt and drought stress. Under extreme temperature stress, MTs protected membrane integrity and also helped plants to adopt multiple strategies such as an enhancement of the number of leaves, side branches, larger primary roots and earliness in flowering. The MT’s response to cold stress is slightly similar to drought and heat as they are involved in minimizing osmotic stress and ion homeostasis; additionally, they exported lipid molecules in order to strengthen the leaf cuticle layer. In case of toxic elements, MTs play a role in the uptake of HMs such as Cd, Co, Cr and Mn and translocated them into root tips and shoot and even to leaves or other upper parts. Moreover, MTs could enhance phytoremediation capacities and also enhanced the activities of antioxidants in order to minimize ROS levels. Taken together, engineering MTs could enhance sugar and water supply, and improved ionic homeostasis could result in increasing PG&D while improving the capacity of plants to withstand dynamic climate changes. Until now, an increasing number of near-to-complete genomes have already been assembled in >400 plant species, and this is still continuing with better versions. Thus, researchers need to explore whole genome MTs as “transportomes” in a sophisticated network related to developmental and environmental cues.

## Figures and Tables

**Figure 1 ijms-22-12792-f001:**
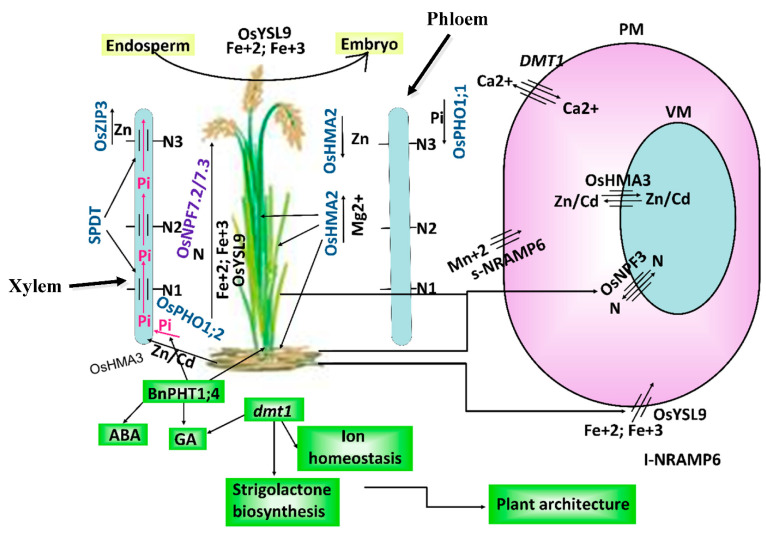
**A thematic model on the role of membrane transporters in improvement of plant architecture, seed yield, and transportation of mineral elements and ion homeostasis.** Figure shows that *OsYSL9* is localized at the plasma membrane (PM) and translocated Fe^+2^ and Fe^+3^ ions from roots to upper plants parts and also exported these ions from the endosperm to embryo, resulting in enhanced grain quality. Several *OsNPF* genes are involved in the transportation of nitrogen towards leaf and, hence, play a role in N use efficiency (NUE). *OsPHO1;2* and *BnPHT1;4* transported the available phosphorus (Pi) from the roots to xylem (xylem loading) and then SPDT export Pi across the nodes. Here, *BnPHT1;4* also upregulate the genes involved in the pathways of ABA, and GA resulted in the improvement of plant growth and regulation. Similarly, *OsHMA3* localized at the vacuolar membrane (VN), which is involved in maintaining Zn/Cd homeostasis and transporting Zn/Cd from root to xylem, and from there onward, the *OsZIP3* gene unloads Zn from xylem to grains. *OsHMA2* is expressed in roots, stem and leaves exported Mg^2+^ across the xylem towards upper parts and translocated Zn in last node towards lower parts across phloem. Lastly, the *OsDMT1* gene localized at the PM and transport Ca^2+^. However, in mutant plants (*dmt1*) upregulation of GA, balancing of ion homeostasis and increased strigolactone biosynthesis processes were observed, which resulted in the overall improvement of plant architecture. For further details about articles used to make the above model, readers can refer to [23,25,26,27,28,29,30,31,32,33,34].

**Figure 2 ijms-22-12792-f002:**
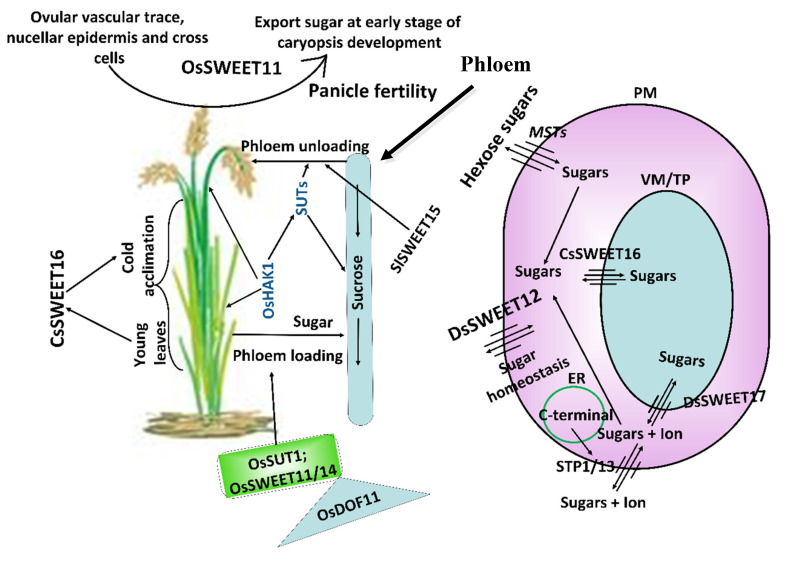
**Sugar transport mechanism in plants.** The figure indicates the localization of *STP1/13*, *DsSWEET12* and *MST* genes at the plasma membrane (PM). The C-terminal motif localized in endoplasmic reticulum (ER) is involved in the PM localization of *STP1/13* genes that transports sugar and ions across the PM. Similarly, *DsSWEET12* and *MSTs* transport only sugar molecules across the PM, and *DsSWEET17* and *CsSWEET16* transport sugar molecules across the vacuolar membrane (VM)/tonoplast (TP). Here, OsDOF11 transcription factor binds with *OsSUT1* and *OsSWEET11/14* and upregulates their expressions levels, which results in phloem loading (transports sugars from leaf photosynthesis apparatus to phloem). *OsHAK1* upregulates downstream SUTs involved in phloem loading (leaf to phloem) and unloading (phloem to panicles) as well as upregulation of *MSTs*. *CsSWEET16* is highly expressed in young leaves and plays a role in cold acclimation. *OsSWEET11* is expressed in reproductive tissues such as ovular vascular trace, nucellar epidermis and cross cells and exports sugar at the early stages of caryopsis development (post-phloem loading), resuling in enhanced grain yield. For more details about the model, readers can refer to [2,12,42,43,44,47,48,49].

**Table 1 ijms-22-12792-t001:** Role of membrane transports in plant growth and development.

Transporter Protein	Plant Species	Localization in Tissue/Organ/Cell	Biological Role	Ref.
OsNPF7.3	*O. sativa*	Lateral roots and stems	Contribute to NUE and grain yield	[23]
OsNPF7.2	*O. sativa*		Enhance tiller number and grain yield	[22]
OsZIP7	*O. sativa*	Parenchyma cells of vascular bundles in roots and nodes	Xylem loading in roots and transfer of Zn/Cd to grain	[24]
OsYSL9	*O. sativa*	Roots and non-juvenile leaves	Distribute iron to developing grains	[25]
OsHMA3	*O. sativa*	Tonoplast/roots	Zn detoxification in roots and storage in vacuoles	[30]
OsDMT1	*O. sativa*		Regulate plant architecture and ion homeostasis	[34]
NRAMP6	*O. sativa*	PM	Transport Fe and Mg and disease resistance	[26]
ZmMGT12	*Z. mays*	Root, stem and leaves	Maintain Mg homeostasis in chloroplast	[36]
ZmSUT2	*Z. mays*	Tonoplast	It acts as sucrose/H^+^ symporter on the vacuolar membrane and remobilizes stored sucrose for subsequent growing tissues	[40]
TaPTR2.1	*T. aestivum*	Tonoplast	Regulates water status during seed germination at early stage	[50]
BnaPHT1	*B. napus*	PM	Pi acquisition and homeostasis and responds to various nutrient stresses including N, K, S and Fe	[39]
BnaPHT1;4	*B. napus*	Cotyledons of early developing seedlings	Pi homeostasis, seed germination and seedling growth through modification in biosynthesis of ABA and GA	[29]
OsZIP7	*A. thaliana*	PM	Increases Zn concentration by 25% in the shoot of transgenic plants	[51]
DsSWEET12	*A. thaliana*	PM	Increased sugar supply and enhanced seedling growth (larger roots and fresh biomass)	[43]
DsSWEET17	*A. thaliana*	Tonoplast	It enhanced root length and fresh weight	[48]
OsDOF11	*O. sativa*	Photosynthetic cells	It upregulated OsSUT1 and OsSWEET11 and 14 genes expression and transported sucrose through apoplastic loading and enhanced resistance against Xanthomonas	[44]
OsSWEET11	*O. sativa*	Ovular vascular trace, nucellar epidermis and cross cells	It remobilizes the sugar from maternal tissues towards maternal–filial interface during early caryopsis developmental stage	[49]
ZmSWEETa/b/c	*Z. mays*	Leaf	Influence on the sugar dynamics from leaves towards developing ears	[52]
AtSUC6	*A. thaliana*	PM	Sugar accumulation in pollen tube and synergid cells	[11]
AtSTP1/13	*A. thaliana*	PM	Involved in sugar transport across cell membranes	[47]
OsHAK1	*O. sativa*	PM	It is involved in controlling vegetative growth, panicle fertility and K^+^ mediated sugar homeostasis	[12]

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
