# Peer review of "The Role of Membrane Transporters in Plant Growth and Development, and Abiotic Stress Tolerance"

_ijms, 2021, doi:10.3390/ijms222312792_

Round 1

Reviewer 1 Report

Novelty. This review manuscript tries to integrate current knowledge about the role of various membrane transporters in plant growth and development as well as in the response to abiotic stresses such as salinity, drought, heat and cold stresses, and heavy metal contamination. The focus is on transporters involved in sink –source interactions, sugar and ionic homeostasis.  Many reviews on membrane transporters have been published on different aspects of plant transporters but not in such an integrative manner.  This is a very ambitious task due to the enormous structural and functional diversity in PTs. The manuscript looks fragmentary in the present form, with lot of irrelevant information especially concerning stresses. It will benefit from improvement.

Some remarks:

Abstract – why only plasma membrane (PM) and tonoplast/ vacuolar membrane (VM) of cells are mentioned? Every membrane-delimitated organelle also contain MTs.

Introduction paragraph lines 63-90. Efficient translocation of mineral elements…..For example OsNPF7.3 MT is involved in the transportation of di-peptides (Gly-68 His) and tri-peptides (Gly-His-Gly) – should be clarified as di/tri-peptides are much more complex compounds than mineral elements. I also wonder how long-distance translocation of methylated arsenic (As) to grains could improve grain quality.

Line 188 – how the authors could explain the fact that NRAMP transporters are involved not only in the ionic balance of Fe and Mn, but also in the response to a fungal pathogen?  This is aside from the central theme.

Lines 215-220 – “Moreover, changes in the accumulation of gibberellic acid (GA) and down-regulation of ABA were also noticed in the transgenic plants. Further, they also showed that exogenous application of GA and ABA were related to increment of seed germination and recovery of advanced seed germination related phenotype, respectively in the over-expressed seeds compared to WT…” – these statements are not related to  the central theme of the review. At least some explanation should be given about the role of MTs in hormonal regulation. Strigolactone for example appears only in Fig. 1.

Line 260 – “accumulation of two-fold various sugars” – unclear

Line 430 – what is SNP?

Lines 467-468 – “play a significant in tackling various posed environmental stresses” – unclear

line 511 – can be utilized

line 535 – how ROS scavenging could be linked to CrPIP2;3 overexpression + drought stress?

Paragraph lines 554-574 Optimization of photosynthesis related tissues is very important for drought tolerance but it is out of the theme “membrane transporters”

Line 610 – uncovered or discovered?

Lines 616-645 – Heat stress response in general is given and only one mention about MTs under heat stress (chloroplast translocons).The next paragraph -  it is not clear if fatty acid desaturases are transporter molecules.

Line 709 – they sated? Unclear

Paragraph lines 693-715 is about cold stress but where is the information about MTs?

The stress part of the manuscript seems like compilation of different facts and not like synthesis and insight about the role of MTs in abiotic stress. Stress related biochemical data should be matched with membrane transporters where it is possible and shortened where is not possible.

Is there any in common comparing PG&D and stress response? Growth inhibition is a well-known stress effect on plants, phenology suffers too.

Maybe a short description of each of the reported kind of MTs should be given before discussing mutants and transcript changes of this particular MT.

The idea of” transportome” exploration is very interesting.

Author Response

Manuscript ID: ijms-1435809

Title: The Role of Membrane Transporters in Plant Growth and Development, and Abiotic Stress Tolerance

Response to reviewer’s Comments to Author

Dear Editor,

Thank you for forwarding the detailed comments of the reviewers which have been helpful to improve our manuscript. We have carefully revised the manuscript according to the suggestions and critical comments. Our response to the reviewers’ comments is given below and also mentioned individual in the MS. All the changes in the MS were colored “Bright Green” to save the time of respected Editor and reviewers. Please feel free to contact us if any problem still exists.

Yours sincerely,

Authors: Rafaqat Ali Gill

General Remarks: Novelty; This review manuscript tries to integrate current knowledge about the role of various membrane transporters in plant growth and development as well as in the response to abiotic stresses such as salinity, drought, heat and cold stresses, and heavy metal contamination. The focus is on transporters involved in sink –source interactions, sugar and ionic homeostasis.  Many reviews on membrane transporters have been published on different aspects of plant transporters but not in such an integrative manner. This is a very ambitious task due to the enormous structural and functional diversity in PTs. The manuscript looks fragmentary in the present form, with lot of irrelevant information especially concerning stresses. It will benefit from improvement.

Response: Thank you for your kind appreciation and also highlighting the weaknesses in our MS. I also acknowledge your time on the detailed overview/comments on our MS. We tried our best to improve the MS and addressed the raised concerns.

Specific Remarks:

Remark 1: Abstract – why only plasma membrane (PM) and tonoplast/ vacuolar membrane (VM) of cells are mentioned? Every membrane-delimitated organelle also contains MTs.

Response: Thank you for the correction. The sentence has been modified in the revised version as “The proteins of membrane transporters (MTs) are embedded with membrane-bound organelles and are prime targets for improvements in the efficiency of water and nutrient transportation”.  

Remark 2: Introduction paragraph lines 63-90. Efficient translocation of mineral elements. For example, OsNPF7.3 MT is involved in the transportation of di-peptides (Gly-68 His) and tri-peptides (Gly-His-Gly)–should be clarified as di/tri-peptides are much more complex compounds than mineral elements.

Response: The sentence has been modified “For example, OsPTR6 MT is involved in the transportation of Gly-His and Gly-His-Gly peptides, showed substrate selectivity for di- and tri-peptides, and its enhanced expression rate increased the growth of rice plants”. Also added an original source of citation [7].

Remark 3: I also wonder how long-distance translocation of methylated arsenic (As) to grains could improve grain quality.

Response: Thank you for the comment. In rice grain, In rice grain, two arsenic (As) species are dominant such as inorganic [mainly, As(III)] and methylated As species (mainly dimethylarsenate, DMA)—methylated by microbial methylation in the soil (vary from 0-80%).A peptide transporter OsPTR7 (OsNPF8.1) uptake of DMA and then translocate (long distance transportation—from roots) to grains but concentration of DMA is dependent on 1) soil condition (variations in the methylation level) and 2) varietal uptake capabilities of rice cultivars (genotypic variation). Compared to plants, for human consumption inorganic As is more toxic than DMA. So, replacement of As(III) with DMA enhanced the grain quality. This above all are detailed in the cited articles [10].  

Remark 4: Line 188 – how the authors could explain the fact that NRAMP transporters are involved not only in the ionic balance of Fe and Mn, but also in the response to a fungal pathogen?  This is aside from the central theme.

Response: Thank you for the comment. In fact, we wrote on the dual role of NRAMP but I also agreed with you as it is against the main theme of this review. So, I removed the pathogen related sentences in the revised version. 

Remark 5: Lines 215-220 – “Moreover, changes in the accumulation of gibberellic acid (GA) and down-regulation of ABA were also noticed in the transgenic plants. Further, they also showed that exogenous application of GA and ABA were related to increment of seed germination and recovery of advanced seed germination related phenotype, respectively in the over-expressed seeds compared to WT…” – these statements are not related to the central theme of the review. At least some explanation should be given about the role of MTs in hormonal regulation. Strigolactone for example appears only in Fig. 1.

Response: Thank you for the comment. We have explained the role of PHT1;4 MT in hormonal regulation as it involved in the upregulation of GA and down-regulation of ABA pathways related genes. Previous literature has also evidenced the antagonistic role of GA and ABA as primary endogenous regulators of seed germination. For instance, GA promote transition of dormancy to germination but the regulatory role of ABA is vice versa [29, 30].

Response: This sentence has been replaced with “two-fold accumulation of various sugars”.

Response: Thank you for the comment. SNP replaced with “single nucleotide polymorphic markers”

Response: Sentence has been modified as “play a significant role in tackling various posed environmental stresses”.

Remark 10: line 535 – how ROS scavenging could be linked to CrPIP2;3 overexpression + drought stress?

Response: Thank you for the comment. Generally, environmental stresses induced ROS that cause the cellular damages resulted in the reduction in the plant growth and development. Contrary, growth enhancers or specifically stress responsive genes mitigate ROS levels in the cell and then finally provides relieves to the plant. However, in this special case CrPIP2;3 improve the water supply/ homeostasis but not mitigate ROS levels. If you still not satisfy then I can delete this angle.

Remark 11: Paragraph lines 554-574 Optimization of photosynthesis related tissues is very important for drought tolerance but it is out of the theme “membrane transporters”,

Response: Thank you for the comment. We have added a new paragraph on the role of MTs in ABA regulation and transportation to other cells under drought stress conditions.

Remark 12: Line 610 – uncovered or discovered?

Response: “uncovered” has been replaced with “discovered”

Remark 13: Lines 616-645 – Heat stress response in general is given and only one mention about MTs under heat stress (chloroplast translocons). The next paragraph - it is not clear if fatty acid desaturases are transporter molecules.

Response: Thank for the comment. Yes, lipids are not the MTs but they facilitate the transportation/cellular movement or in other words cell initial response to heat stress mechanism is regulated by lipids in the PM. So, we tried to establish a story about the lipid accumulation with an example of “translocon” in this paragraph. In the next paragraph (modified), we tried to establish a story about the role of lipids fluidity and integrity (of PM) in the Ca+2 influx (i.e., transported through CNGC6 and binding with CaMs) towards PM. In a nutshell, in the both paragraphs, lipid accumulation, fluidity and integrity in the PM facilitated the Ca+2 response and signalling that triggered the “systemic acquired acclimatization” of plants by activation HSFs and HSPs in a sophisticated network. Please see revised paragraph, line 674-713.

Response: “They sated” has been replaced with “They stated”.

 Remark 15: Paragraph lines 693-715 is about cold stress but where is the information about MTs?

Response: Thank you for the comment. In this paragraph story was established to showed the identification of QTLs harbouring genes coding for MTs from the big data though association mapping and other genomic approaches. In this case, they find a MT gene coding UDP-glucosyltransferase (a super family of membrane-bound enzymes that involved in the glycosylation of endoplasmic reticulum (ER) localized additional glycoproteins. Here, it acts as a MT and transfer the sugar molecules across EM membranes. Additionally, it play a pivotal role in regulating the various activities of flavonoids including stability, availability and biological activities [93, 94]) and detailed its role in sugar transportation under the cold stress environment. I added these above lines related to UDP-glucosyltransferase as per your kind suggestions and cited more relevant articles.

Remark 16: The stress part of the manuscript seems like compilation of different facts and not like synthesis and insight about the role of MTs in abiotic stress. Stress related biochemical data should be matched with membrane transporters where it is possible and shortened where is not possible.

Response: Thank you for the comment. I agreed there were shortcomings specially in the drought and heat stress parts. So, in the revised version I added one long paragraph in drought (line 555-600) and modified the whole paragraph in heat stress (line 674-713) and tried to add more relevant information.  

Remark 17: Is there any in common comparing PG&D and stress response? Growth inhibition is a well-known stress effect on plants, phenology suffers too.

Response: Thank you for the comment. We focus on the role of MTs in PG&D and Abiotic stresses perspectives but not the comparison angle—that is a great idea and can see in the future article. As we already discussed three sub-headings in PG&D part and five in the abiotic stress part.

Remark 18: Maybe a short description of each of the reported kind of MTs should be given before discussing mutants and transcript changes of this particular MT.

Response: Thank you for the comment. A short description of MTs has been highlighted. Please see lines 146, 158, 171, 192, 202, 263, 276, 380, 411, 443, 454, 468, 518, 614, 628, 699, 723, 758, 828, 843, 851, 885. For more details about MTs (protein name, plant species, cellular localization and biological role etc.) are presented in the table 1 and 2.  

Remark 19: The idea of” transportome” exploration is very interesting.

Response: Thank you for the kind appreciation.

Thank!

Reviewer 2 Report

While the factual information in this review may be of interest to specialists in this specific topic, general readers may find it more interesting if more critical view of the research area and future perspectives are clearly stated in each section. The article tries to show too much. The narrative is too grand. The section on plant growth and development is not sufficiently related to the section on abiotic stress tolerance, and additional literature is needed to link the two sections. 

The molecular mechanisms of MT involved in plant growth and development and abiotic stress tolerance are too briefly described and need to be reviewed and summarized in more literature.

The relationship between the part of 2.2 and 2.3 and plant growth and development is not fully demonstrated.

The abbreviated form of the word in the abstract, which should be uniform in the later text.

too much writing mistakes:

L53-54 are not sufficiently linked to the theme of the article and to the context

Line 107-108  using Different tenses。

Line 120 “try to provides”

Line 135 NPF7.2 is a nitrate transporter gene

Line 147 expression - the expression

Line 149 reduction - a reduction

Line 151 was - were

Line 178-179 although… but…?

“NRAMP(Natural resistance-associated macrophage proteins)“ should be written in the same uniform as the other words, ”Natural resistance-associated macrophage proteins(NRAMP)”.

Line 220-222 some mistakes.

Line 69 if “the above MT” means “OsNPF7.3”, “respond” and “mediate” should add “s/ed”.

Line 136-139 “In detail…” the sentence is not clear, please correct and rewrite it.

Line 194-196 “involvement” how? Please write it clearly.

Line 208 Do you have reference to support this point?

Line 425 “HKT MT”?

Author Response

General Remark: While the factual information in this review may be of interest to specialists in this specific topic, general readers may find it more interesting if more critical view of the research area and future perspectives are clearly stated in each section. The article tries to show too much. The narrative is too grand. The section on plant growth and development is not sufficiently related to the section on abiotic stress tolerance, and additional literature is needed to link the two sections.

Response: Thank you for your kind appreciation and critical comments that surely helped us to further improve our MS. The idea of relationship/comparative angle on the role of MT in PG&D and abiotic stresses is great (but in my humble opinion the MS will be lengthier as we have to connect three sub-heading of PG&D with five headings of abiotic stress parts) and we shall see in the future article. If you still not satisfied then we can try to add this angle too. 

Specific Remarks:

Remark 1: The molecular mechanisms of MT involved in plant growth and development and abiotic stress tolerance are too briefly described and need to be reviewed and summarized in more literature.

Response: Thank you for the comment. I agree with you especially in cases of drought and heat stresses. In the revised version in each section, we have added two long paragraphs (drought stress lines 560-605 and heat stress 679-718) and tried to provide more evidence in an easy-to-understand manner for the readers.

Remark 2: The relationship between the part of 2.2 and 2.3 and plant growth and development is not fully demonstrated.

Response: Thank you for the comment. We have tried to build a story on the relationship of section 2.2 with PG&D as “Ionic homeostasis of mineral elements by either uptake or through detoxification in the cells is important for the steady PG&D. For instance, HMAs maintain Zn (lines 174-183) level in the root. NRAMPs (lines 183-194) are involved in the transportation of Fe and Mn that resulted in the increased biomass production. MGT transport Mg+2 in different cell organs/tissues (from root-leaves) and help chlorophyll synthesis process takes place in chloroplast (in the leaves). PHT transporter involved in the transportation of P. The over-expression of PHT1:4 induced the seed germination and enhance the seedling growth directly and through hormonal regulation—enhance expression levels of genes related to GA and down-regulated the genes involved in the ABA pathways. Similarly, we tried to establish the interaction of section 2.3 and PG&D started with insight on role of SUT (see lines 268-280). We reviewed its role on remobilization of sucrose from vacuoles and then transport to plant’s growing tissues to enhance the biomass and kernel yield. From lines (281-304), here we insight on the dominant sugar transport family SWEET and argued that DsSWEET12 involved in increment of root length and fresh biomass in continuous sugar-supply-dependent manner. Further, we discussed on the role of OsDOF11 (lines 304-320) that acts as a transcription factor, which increases the gene expression levels of other MTs such as OsSWEET11/14 and OsSUT1. In mutant osdof1 plants, root length of young seedlings was reduced due to less sugar supply. I hope, now the relationships will clearer to you.        

Remark 3: The abbreviated form of the word in the abstract, which should be uniform in the later text.

Response: Thank you for the comment. We used the abbreviations i.e., MT, PG&D, HMs, PM and VM in the abstract tried to made sure the uniformity in the MS too.

Remark 4: Too much writing mistakes: L53-54 are not sufficiently linked to the theme of the article and to the context.

Response: Thank you for highlighting mistakes in our MS. We have tried to make connection with theme as “To combat with food insecurity to feed the growing population, there is need to design studies on whole genome membrane transporters (MT) for proper utilization of the plant energy under dynamic environmental changes to produce high yielding crops”.

Remark 5: Line 107-108 using Different tenses,

Response: The sentence has been modified as “Over-expression of TaZnF in A. thaliana significantly increased tolerance to heat and cold stress as well as promoted early flowering”.

Remark 6: Line 120 “try to provides”,

Response: The sentence has been modified as “Lastly, we have tried to provide the recent understandings how MTs respond to changing environments, for instance, strengthening of leaf sheath and increased sugar supply”.

Remark 7: Line 135 NPF7.2 is a nitrate transporter gene,

Response: The sentence has been modified as “For example in rice, OsNPF7.2 is a nitrate transporter gene, which is responsible in regulating the tiller number and grain yield”.

Remark 8: Line 147 expression - the expression, Line 149 reduction - a reduction, Line 151 was – were and Line 178-179 although… but…?, Line 220-222 some mistakes.

Response: Thank you for the corrections. These suggested corrections have been added in the revised version.

Remark 9: “NRAMP(Natural resistance-associated macrophage proteins)“ should be written in the same uniform as the other words, ”Natural resistance-associated macrophage proteins(NRAMP)”.

Response: Thank you for the comment. We have changed as suggested “Natural Resistance-Associated Macrophage Proteins (NRAMPS)”.

Remark 10: Line 69 if “the above MT” means “OsNPF7.3”, “respond” and “mediate” should add “s/ed”.

Response: Thank you for the suggestion. We have modified the sentence as “In another study, OsNPF7.3 MT responds to organic nitrogen (N) supply and mediated N use efficiency (NUE) that resulted in the increased number and filled panicles per plant, N content in grain and grain yield”.

Remark 11: Line 136-139 “In detail…” the sentence is not clear, please correct and rewrite it.

Response: Thank you for the kind comment. We have modified the entire sentence asFurther, their results showed that in the over-expressed lines tiller number and seed yield were significantly higher compared with mutant lines”.

Remark 12: Line 194-196 “involvement” how? Please write it clearly.

Response: Thank you for the comment. We have simplified the sentence as “Moreover, in rice mutant nramp6 plants, a significant deterioration of biomass was recoded that indicated its importance in PG&D”.

Remark 13: Line 208 Do you have reference to support this point?

Response: Thank you for highlighting the point. I speculated based on above mentioned finding as “The expression level of ZmMGT12 were higher in roots, stems and leaves but was more prominent in leaves. Interestingly, higher expression rate in leaves was dependent on the rate of chlorophyll synthesis”. As expression level of ZmMGT12 is dependent on the rate of chlorophyll synthesis—indicated that ZmMGT12 play a role in Mg+2 transportation from root-via-stem to leaves (where it highly expressed). I have modified the sentence as “Taken together, these findings suggested that ZmMGT12 was responsible for transportation of Mg+2 ion and probably to the chloroplast in the leaves”.

Remark 14: Line 425 “HKT MT”?

Response: Thank you for the comment. Replaced “HKT MT” with “high-affinity potassium transporter (HKT)”.

I am highly thankful to the Reviewer_2 for their time that really helpful to further improve our MS. Please feel free to ask if anything still not clear or need more explanation.

Regards!

Round 2

Reviewer 1 Report

The manuscript has been improved but still needs some revision, especially the English. Some remarks:

Abstract - MTs, PG&D, HM, PM and VM - without definitions; better use the whole words instead of abbreviations;   embedded in

Introduction – could be shortened, too many details which are repeated in the other sections

Line 130 has recently been studied…. And has also been genetically verified in….

Line 236 – were expressed

Line 300 – had longer roots

Line 302 – were dependent

Line 303 - can further enhance their diverse roles

Line 329 – and were more expressed when

Line 338 - root is always remained

Line 386 – that are involved

Line 400 - indicated that how precisely MTs are responsible for K+ uptake

Line 406 - Ideally, to get insight on

Line 427 - genotypes that showed an increase in … can be incorporated

Line 432 - maize varieties were collected

Line 434 – environment, which resulted in the forms/in the above-mentioned

Line 487 - Detailed investigation

Line 494 – focused

Line 502 - well-utilize …… can bring

Line 506 – what is SNP - single nucleotide polymorphism?

Line 537 -  to evaluate

Line 561-606 – how the paragraph of hormonal homeostasis is related to source-sink interactions? I think it should be omitted.

Line 623 - it is starts expression – its expression starts

Line 631 - overall improvement

Line 688 - the content of polyunsaturated and polyunsaturated fatty acids

Line 739 - that made of lipids/ or /that is made of lipids

Line 744 - tress tolerant plant

Line 778 - flavonoid biosynthesis resulting in the accumulation

Line 786 - To get insight on cold sensing

Line 806 - PM physiology is dependent on

Line 847 – detoxified, otherwise they can deteriorate

Line 876 - acts as Cd extrusion pump contributing to

Line 924 - of all four metals were varied in all organs but more prominently in roots

Line 935 - sequencing of many plant genomes thereby opening

Line 249 - Table 2 and Figure 3 describe

Lines 963-968 – are these sentences continuation of the text below fig 3?

Line 985 - coordination is disturbed

Line 994 - such as enhancement of

Line 1002 – Until now

Author Response

Manuscript ID: ijms-1435809

Title: The Role of Membrane Transporters in Plant Growth and Development, and Abiotic Stress Tolerance

Response to reviewer’s Comments to Author

Dear Editor,

Thank you for forwarding the detailed comments (R2) of reviewer 1, which have been helpful to improve our manuscript. We have carefully revised the manuscript according to the suggestions (including English grammatical issues) and critical comments. Our response to the reviewers’ comments is given below and also mentioned individually in the MS. All the changes in the MS were colored “Bright Green” to save the time of the respected Editor and reviewer 1. Please feel free to contact us if any problem still exists.

Yours sincerely,

Authors: Rafaqat Ali Gill

General Remarks: The manuscript has been improved but still needs some revision, especially the English.

Response: Thank you for your appreciation of our response to revision and again spent time on our MS. We really thank you for helping us to further improve our MS.

Specific remarks:

Remark 1: Abstract - MTs, PG&D, HM, PM and VM - without definitions; better use the whole words instead of abbreviations; embedded in

Response: Abbreviation used after adding the full name. Those who used only one time added their full names only.

Remark 2: Introduction – could be shortened, too many details which are repeated in the other sections

Response: Thank you for the comment. We have modified line 68, cut lines 70-71 and 75-78 in the second paragraph of the introduction as it looks repetitive to section 2. Also, we tried to modify section 2.1 lines 151-154 accordingly.

Remark 3: Line 130 has recently been studied…. And has also been genetically verified in….

Line 236 – were expressed

Line 300 – had longer roots

Line 302 – were dependent

Line 303 - can further enhance their diverse roles

Line 329 – and were more expressed when

Line 338 - root is always remained

Line 386 – that are involved

Line 400 - indicated that how precisely MTs are responsible for K+ uptake

Line 406 - Ideally, to get insight on

Line 427 - genotypes that showed an increase in … can be incorporated

Line 432 - maize varieties were collected

Line 434 – environment, which resulted in the forms/in the above-mentioned

Line 487 - Detailed investigation

Line 494 – focused

Line 502 - well-utilize …… can bring

Line 506 – what is SNP - single nucleotide polymorphism?

Line 537 -  to evaluate

Line 623 - it is starts expression – its expression starts

Line 631 - overall improvement

Line 688 - the content of polyunsaturated and polyunsaturated fatty acids

Line 739 - that made of lipids/ or /that is made of lipids

Line 744 - tress tolerant plant

Line 778 - flavonoid biosynthesis resulting in the accumulation

Line 786 - To get insight on cold sensing

Line 806 - PM physiology is dependent on

Line 847 – detoxified, otherwise they can deteriorate

Line 876 - acts as Cd extrusion pump contributing to

Line 924 - of all four metals were varied in all organs but more prominently in roots

Line 935 - sequencing of many plant genomes thereby opening

Line 949 - Table 2 and Figure 3 describe

Line 985 - coordination is disturbed

Line 994 - such as enhancement of

Line 1002 – Until now

Response: Thank you very much for the detailed overview of our MS and the corrections. These all suggestions have been added and highlighted in the MS. Also, we re-checked the whole MS and tried to remove such grammatical mistakes in the revised MS.

Remark 4: Lines 963-968 – are these sentences continuation of the text below fig 3?

Response: Yes, these lines are part of figure 3 legend. I changed the format (wider) same as the above-mentioned figure 3 legend.

Remark 5: Line 561-606 – how the paragraph on hormonal homeostasis is related to source-sink interactions? I think it should be omitted.

Response: In this paragraph, we tried to establish a story about the role of MTs in the transportation of ABA from source to sink organs/tissues in ATP and drought stress-dependent manner. In the first case, AtABCG25 plays a role in ABA transportation in ATP dependent manner, and in over-expressed plants showed higher stomatal regulation compared with WT. In the second case, we discussed the pivotal role of AtABCG40 that it enhances ABA uptake, upregulates the ABA-responsive genes and quick stomatal closure resulted in improved drought tolerance. Similarly, AtABCG22 expressed higher in guard cells and is required for the regulation of stomatal opening/closure under drought stress. Also, it is involved in lowering the leaf temperature and improving the transpiration rate/water loss. More importantly, we discussed the double mutants atabcg22 (defective of ABA signalling) plants, which showed enhanced water loss and changes in the overall plant phenotypes of other mutant nced3 (defective in ABA biosynthesis) plants compared with WT. Thus, AtABCG22 plays a dual additive function both in ABA signalling and biosynthesis. As a whole, the above-mentioned MTs play a significant role in ABA transportation, remobilization, homeostasis, and biosynthesis in sink dependent manner under drought stress. 

This manuscript is a resubmission of an earlier submission. The following is a list of the peer review reports and author responses from that submission.

Round 1

Reviewer 1 Report

The MS entitled "The Role of Membrane Transporters in Plant Growth and Development, and Abiotic Stress Tolerance" is a modified version of many recently published systematic reviews. Based on the critical evaluation of the contents of the MS the following suggestions are furnished:

1. The MS mostly lacks novelty. Compilation of dataset without directing it into a systematic framework is an immense flaw in this MS. The sheer lack of foresight for casting the collected information led to enormous mutilation. I would strongly recommend dismantling the sections and rearrange the information in a methodological pattern.

2. The second most baffling aspect of this MS was that it was devised in an incongruous fashion. Again, I would compel the authors that the contents should be simplified and make then comprehensible.

3. More importantly, I have noticed that the authors have largely rearranged information in Table 1, Figures 1 and 2 without citing the original source from where the concept/ design was adopted. Also, some of the most recent citations on the subject are missing though information from the same was included in the MS.

4. Lastly, the claims made by the authors in the conclusion section have been largely extrapolated, pruning them to match in a realistic situation is advisable.

Author Response

General Comments: The MS entitled "The Role of Membrane Transporters in Plant Growth and Development, and Abiotic Stress Tolerance" is a modified version of many recently published systematic reviews. Based on the critical evaluation of the contents of the MS the following suggestions are furnished:

Specific comments:

Remark 1. The MS mostly lacks novelty. Compilation of dataset without directing it into a systematic framework is an immense flaw in this MS. The sheer lack of foresight for casting the collected information led to enormous mutilation. I would strongly recommend dismantling the sections and rearrange the information in a methodological pattern.

Answer: Thank you for the comment, however, with due respect we disagree.

Remark 2. The second most baffling aspect of this MS was that it was devised in an incongruous fashion. Again, I would compel the authors that the contents should be simplified and make then comprehensible.

 Answer: We have tried our best to present our MS based on the recently published research/review (mainly research) articles and approximately more than 80 % from last four years.

Remark 3. More importantly, I have noticed that the authors have largely rearranged information in Table 1, Figures 1 and 2 without citing the original source from where the concept/ design was adopted. Also, some of the most recent citations on the subject are missing though information from the same was included in the MS.

 Answer: Thank you for the comment. We have cited the articles in the individual figure legend based on which we made the models. Please have a look on the revised figure legends in the MS.

Remark 4. Lastly, the claims made by the authors in the conclusion section have been largely extrapolated, pruning them to match in a realistic situation is advisable.

Answer: The claim in the conclusion section was made based on recent articles (cited in the main text and also in conclusion section). However, numbers of the MTs in B. napus has been modified (after re-check from the original source).

Thanks!

Reviewer 2 Report

The authors have written a review article on the Role of Membrane Transporters in Plant Growth and Development, and Abiotic Stress Tolerance. Authors described the role of MT and discussed the gene expression level changes and genomic variations within a species as well as within a family in response to developmental and environmental cues.

  1. Figure 1 and Figure 2 could improved with better quality.
  2. Genes list with NCBI accession numbers for different membrane transporters from other plants studied for the Plant Growth and Development, and Abiotic Stress Tolerance could be included as additional file. This information will be really useful for other researchers.

Author Response

General remark: The authors have written a review article on the Role of Membrane Transporters in Plant Growth and Development, and Abiotic Stress Tolerance. Authors described the role of MT and discussed the gene expression level changes and genomic variations within a species as well as within a family in response to developmental and environmental cues.

Specific remarks:

Remark 1: Figure 1 and Figure 2 could improve with better quality.

Answer: Thank you for the suggestion. We have tried our best to improve the quality further.

Remark 2: Genes list with NCBI accession numbers for different membrane transporters from other plants studied for the Plant Growth and Development, and Abiotic Stress Tolerance could be included as additional file. This information will be really useful for other researchers.

Answer: Thank you for the suggestion. Yes, I agreed for the benefit of mass readers it will be helpful especially for working in areas of genome-wide/wet lab verification studies. We have prepared the supplementary file containing NCBI_accession_IDs corresponding to gene_IDs mentioned in Table 1 and Table 2.

Thanks!

Reviewer 3 Report

Review manuscript entitled "The Role of Membrane Transporters in Plant
Growth and Development, and Abiotic Stress Tolerance" is submitted by Gill and co-authors. It described the important role of MTs in growth, development, and abiotic stress tolerance. They provide an up to date information on MTs and their role in development and stress. Tables and figures are well designed with lots of information.  Therefore, I would like to recommend this paper for further process. However, the reviewer has some suggestions and are as follow:

 Line 81- AtSUC7 (Italic)

Line 99-In response, plants activate Na + /K + MTs to achieve the homeostasis. Which one? Kindly mentioned the name of MTS.

Line 225- tal of 49 PHT1 gene (Italic)

Line 227- 27 PHT1 gene (Italic)

Line 230- PHT1 gene (Italic)

Figure 1. Thematic model represented the role of membrane transporters in improvement of plant 241 architecture, seed yield and transportation of mineral elements and ion homeostasis. Kindly write in details about what is going on in the image.

Line 245- OsNPF gene (Italic)

Line 246 wards leaf and involved in N use efficiency (NUE).

Line 253- transport Zn/Cd from root to xylem and from onward OsZIP3 gene (Italic)

Line 255- stem and leaves export Mg+2 Mg2+ across xylem towards upper parts and

Line 257-Lastly, DMT1 gene (Italic) localized at the PM and transport Ca+2 (Ca2+)

Line 258-in the mutant plants (dmt1) upregulation of GA, balancing of ion ho

Line 271- nels was recorded in mutant (zmsut2) plants compared to wild type 272 (WT). Moreover, in zmsut2 mutants, accumulation of two -fold vari

Line 609- 609 reported that loss of function mutation (ala6) resulted in the sensitive

Line 614-till heat susceptibility was noticed just like loss -of -function ala6 mutant. Moreover, in ala6 mutant plants

Line 641-toc33 mutant displayed accumulation of HSP70 protein, while toc66

Line  802-PtABCC1, a member of ABC transporter was cloned from Populus trichocarpa and-Why all sentence is italic?

Line 837-TaHMA2 derivative (Non-italic)

Figure 3-Kindly write detailed legends so that easily can understand the figure.

Author Response

General remarks: Review manuscript entitled "The Role of Membrane Transporters in Plant Growth and Development, and Abiotic Stress Tolerance" is submitted by Gill and co-authors. It described the important role of MTs in growth, development, and abiotic stress tolerance. They provide an up to date information on MTs and their role in development and stress. Tables and figures are well designed with lots of information.  Therefore, I would like to recommend this paper for further process. However, the reviewer has some suggestions and are as follow:

Answer: Thank you for kind appreciation.

Specific remarks:

Remark 1: Line 81- AtSUC7 (Italic), Line 99-In response, plants activate Na + /K + MTs to achieve the homeostasis. Which one? Kindly mentioned the name of MTS. Line 225- tal of 49 PHT1 gene (Italic), Line 227- 27 PHT1 gene (Italic), Line 230- PHT1 gene (Italic).

Answer: Thank you for the comments. We have incorporated the said suggestions in the revised MS.

Remark 2: Figure 1. Thematic model represented the role of membrane transporters in improvement of plant architecture, seed yield and transportation of mineral elements and ion homeostasis. Kindly write in details about what is going on in the image.

Answer: Thank you for kind suggestion. We have written the figure legend with more details in the revised MS.

Remark 3: Line 245- OsNPF gene (Italic), Line 246 wards leaf and involved in N use efficiency (NUE). Line 253- transport Zn/Cd from root to xylem and from onward OsZIP3 gene (Italic) Line 255- stem and leaves export Mg+2 Mg2+ across xylem towards upper parts and Line 257-Lastly, DMT1 gene (Italic) localized at the PM and transport Ca+2 (Ca2+), Line 258-in the mutant plants (dmt1) upregulation of GA, balancing of ion ho.., Line 271- nels was recorded in mutant (zmsut2) plants compared to wild type 272 (WT). Moreover, in zmsut2 mutants, accumulation of two -fold vari…, Line 609- 609 reported that loss of function mutation (ala6) resulted in the sensitive Line 614-till heat susceptibility was noticed just like loss -of -function ala6 mutant. Moreover, in ala6 mutant plants, Line 641-toc33 mutant displayed accumulation of HSP70 protein, while toc66, Line  802-PtABCC1, a member of ABC transporter was cloned from Populus trichocarpa and-Why all sentence is italic? Line 837-TaHMA2 derivative (Non-italic)

Answer: Thank you for the detail look at our MS. We have revised our MS accordingly.

Remark 4: Figure 3-Kindly write detailed legends so that easily can understand the figure.

Answer: Thank you for the suggestion. We have written the detailed legend of Figure 3 in the revised version.

Please feel to ask if any ambiguity still exists.

Thanks! 

Round 2

Reviewer 1 Report

Thank you for your response to reviewer's comments and suggestions. The response to comments 1, 2, and 4 are unsatisfactory and the revised manuscript does not adhere to directions presented earlier.